# Speedy A governs non-homologous XY chromosome desynapsis as a unique prerequisite for XY loop-axis organization

Dongteng Liu [1,2,✉], Yuxiang Zhang [3,7], Dongliang Li [1,2,7], Binjie Jiang [2,7], Xudong Zhao [1,2,7], Yanyan Li[1], Zexiong Lin[1,2], Yu Zhao[1], Zhe Hu[2], Shuzi Deng[1,2], Zheng Li[3], Haonan Lu[2], Karen K L Chan[2], William S B Yeung [1,2], Philipp Kaldis [4], Chencheng Yao[3], Hengbin Wang[5], Louise T Chow [6,✉] & Kui Liu [1,2,✉]

## Abstract

In mouse early pachytene spermatocytes, the X and Y chromosomes undergo rapid non-homologous (NH) synapsis and desynapsis, but the functional significance remains unknown. Here, we report that pachynema-specific knockout of Speedy A (SpdyA) from telomeres caused persistent *Y-X NH synapsis*, with the entire Y axis synapsed onto the X axis. This persistent *Y-X NH synapsis* did not interrupt meiotic sex chromosome inactivation, recombination, or sex body formation, but it disrupted X-Y loop-axis organization and homologous X-Y desynapsis, leading to spermatocyte death. Similarly, persistent *Y-X NH synapsis* was also observed in pachytene spermatocytes lacking TRF1, where SpdyA was frequently lost from the X-Y non-pseudoautosomal region (non-PAR) telomeres. Mechanistic studies revealed that Serine 48 of SUN1 is a key SpdyA/CDK2 phosphorylation site required for Y-X NH desynapsis. We propose that SpdyA governs Y-X NH desynapsis by stabilizing the linkage between the X-Y non-PAR telomeres and their LINC complexes, and that this process is regulated independently from other aspects of pachynema progression. Our findings suggest a key role for Y-X NH desynapsis in establishing proper X-Y loop-axis organization.

**Keywords** SpdyA; Telomeres; Non-homologous XY Desynapsis; XY Loop-axis Organization; Mouse Pachytene Spermatocytes
**Subject Categories** Cell Cycle; Development; DNA Replication, Recombination & Repair

## Introduction

In mouse spermatocytes, meiotic prophase I is a relatively lengthy stage where homologous chromosomes undergo pairing, synapsis, and recombination (Baudat et al, 2013; Zickler and Kleckner, 1999). Following synapsis, spermatocytes enter a prolonged substage of pachynema, where Meiotic Sex Chromosome Inactivation (MSCI), crossover formation, and the development of the sex body are accomplished. The X and Y chromosomes share a short region of homology (~0.7 Mb) known as the pseudoautosomal region (PAR) where X-Y recombination occurs (Kauppi et al, 2012). X-Y pairing occurs at the zygonema-to-pachynema transition and is mediated by SPO11-induced double-strand breaks (DSBs) in the PAR at late zygonema (Kauppi et al, 2011). Recent studies have advanced our understanding of how X-Y chromosomes achieve efficient DSB formation for recombination (Acquaviva et al, 2020; Boekhout et al, 2019; Giannattasio et al, 2023; Lampitto and Barchi, 2024; Papanikos et al, 2019). First, the PAR in mouse spermatocytes undergoes a prominent rearrangement of its loop-axis structure, with DNA in the PAR being packaged into many smaller chromatin loops and the axes becoming disproportionately long relative to DNA length compared to autosomes (Acquaviva et al, 2020; Kauppi et al, 2011). This higher-order structure of the PAR potentially promotes a high frequency of DSB formation (Acquaviva et al, 2020; Kauppi et al, 2012). Second, the sister axes of the PARs in X and Y chromosomes undergo a distinctive splitting, and then the SPO11-auxiliary RMMAI proteins (REC114, MEI4, MEI1, ANKRD31 and IHO1) are recruited to the *cis*-acting mo-2 minisatellites on the PAR (Acquaviva et al, 2020; Boekhout et al, 2019; Papanikos et al, 2019). As a result, with the concomitant expression of SPO11β and SPO11α isoforms, DSB formation is boosted in the PAR, leading to X-Y recombination (Giannattasio et al, 2023; Lampitto and Barchi, 2024).

[1]Shenzhen Key Laboratory of Fertility Regulation, Reproductive Medicine Center, The University of Hong Kong-Shenzhen Hospital, Shenzhen, Guangdong, China. [2]Department of Obstetrics and Gynaecology, Li Ka Shing Faculty of Medicine, The University of Hong Kong, Hong Kong SAR, China. [3]Department of Andrology, Center for Men's Health, Department of ART, Institute of Urology, Urologic Medical Center, Shanghai Key Laboratory of Reproductive Medicine, Shanghai General Hospital, Shanghai Jiao Tong University School of Medicine, Shanghai, China. [4]Department of Clinical Sciences, Lund University, Clinical Research Centre (CRC), Malmö, Sweden. [5]Department of Internal Medicine, Division of Hematology, Oncology and Palliative Care, Massey Cancer Center, Virginia Commonwealth University, Richmond, VA, USA. [6]Department of Biochemistry and Molecular Genetics, University of Alabama at Birmingham, Birmingham, AL, USA. [7]These authors contributed equally: Yuxiang Zhang, Dongliang Li, Binjie Jiang, Xudong Zhao.
✉E-mail: ldt88@163.com; ltchow@uab.edu; kliugc@hku.hk

It has long been noticed that the X and Y chromosomes undergo dynamic configurational changes during pachynema (Acquaviva et al, 2020). As illustrated in Fig. EV1, during pachynema in mouse spermatocytes, the X and Y chromosomes first undergo a non-homologous (NH) synapsis with synaptonemal complex (SC) built between the entire (or nearly entire) Y axis and the X axis (Fig. EV1A-a). However, this NH X-Y synapsis is transient and is promptly subjected to NH desynapsis, and the SC between the X and Y axes is gradually shortened (Fig. EV1A-b), and by mid-pachynema the SC is restricted to the PAR (Fig. EV1A-c). By late pachynema, crossover occurs and chiasma is formed between the X and Y chromosomes, followed by homologous desynapsis of the X and Y chromosomes at the PAR, leading to an "end-to-end" attachment of the X and Y chromosomes (Fig. EV1A-d). These changes in the X-Y configuration have been observed in various mammalian species, including humans (Chandley et al, 1984; Solari, 1988; Tres, 1977; Tres, 1979), but the regulatory machineries and physiological significance of such dynamic X-Y configurational changes remain unclear.

Speedy A (SpdyA) is a non-canonical activator of cyclin-dependent kinases (CDKs) and was first reported in *Xenopus* oocytes (Lenormand et al, 1999). Later, SpdyA was found to be localized at chromosome telomeres in prophase I germ cells in both male and female mice, and it is indispensable for tethering telomeres onto the nuclear envelope (NE) at zygonema (Mikolcevic et al, 2016; Tu et al, 2017), which is an essential step for homology searching, recombination, and synapsis (Scherthan, 2007). We and others have reported that knockout of the *Spdya* gene in mice disrupts the telomere-NE attachment and interferes with synapsis, leading to zygonema arrest in spermatocytes and oocytes and subsequent infertility in mice (Mikolcevic et al, 2016; Tu et al, 2017). It is worth mentioning that telomeres are attached to the NE via multiple proteins, including components of the LINC (linker of nucleoskeleton and cytoskeleton) complex, SUN1 (Sad1 and UNC84 domain-containing 1) and KASH5 (KASH domain-containing 5) as well as TERB1 (telomere repeat-binding bouquet formation protein 1), TERB2, and MAJIN (membrane anchored junction protein), which form the TTM complex [for reviews and papers, see (Burke, 2018; Wang et al, 2022)]. The telomere shelterin component TRF1 (telomeric repeat-binding factor 1) can directly bind to SpdyA to protect telomeres from fusion (Wang et al, 2018).

In mouse pachytene spermatocytes, SpdyA localizes to telomeres as well as to the unsynapsed region of X-Y axes (Fig. EV1B). In addition, axial localization of SpdyA is also observed at the unsynapsed region of late-synapsing autosomes in late zygotene spermatocytes (Fig. EV1C). In this study, we found that knockout of SpdyA in early pachytene mouse spermatocytes caused a persistent NH synapsis of X and Y chromosomes with the entire Y axis synapsed onto the X axis, which to our surprise did not disrupt general pachynema events such as MSCI, crossover formation, and sex body formation. This *full Y-X NH synapsis* led to disrupted X-Y loop-axis organization in the sex body, abnormal ubiquitination on sex chromosomes, and eventually non-occurrence of X-Y desynapsis at late pachynema, leading to cell death. Our further analyses of a hypomorphic-SpdyA mouse model showed that the X-Y chromosomes were more vulnerable to reduced telomeric SpdyA than autosomes, which easily suffer from

*full Y-X NH synapsis*. Moreover, our mechanistic analyses revealed that SpdyA initiates the Y-X NH desynapsis by stabilizing the linkage between the non-PAR telomeres of X-Y and the LINC complex at early pachytene. The Y-X NH desynapsis is emphasized in this study as a prerequisite for proper X-Y loop-axis organization in the sex body and the eventual X-Y desynapsis, which is essential for preventing the generation of sperms with X-Y aneuploidy.

# Results

## Specific deletion of the *Spdya* gene in mouse pachytene spermatocytes

To study the specific functions of SpdyA in pachytene spermatocytes, we generated a pachynema-specific knockout of the *Spdya* gene using the *Spdya^{fl/fl};Ddx4-Cre^{ERT2}* mice (Fig. 1A). Briefly, two consecutive intraperitoneal (i.p.) injections of tamoxifen (20 mg/kg body weight) were administered to male mice at postnatal day (PD) 16 and PD17, and testes were collected 5 days post-injection. As shown in Fig. 1B, compared to control *Spdya^{fl/fl}* spermatocytes, where SpdyA protein localized to telomeres (Fig. 1B, a–a″, arrowheads) and sex chromosome axes (Fig. 1B-a′, arrows), the conditional *Spdya*-knockout (*Spdya^{cKO}*) spermatocytes exhibited a depletion of SpdyA signals at telomeres (Fig. 1B, b–b″, arrowheads) and sex chromosome axes (Fig. 1B-b′, arrow). At the same time, the *Spdya^{cKO}* cells displayed normal pachytene SYCP1 (synaptonemal complex protein 1) signals on autosomes (Fig. 1B, b–b″) and aggregated γH2AX (histone variant H2AX phosphorylated at serine 139) signals in the sex body area (Fig. 1B-b, dashed circle). Quantitative analysis revealed that 68.8% of pachytene spermatocytes in tamoxifen-treated *Spdya^{fl/fl};Ddx4-Cre^{ERT2}* mice (n = 229) were *Spdya^{cKO}* pachytene cells (Fig. 1B-c).

## Deletion of the *Spdya* gene resulted in persistent *full Y-X NH synapsis* in pachytene spermatocytes

In mice, centromeres are located near one end of the chromosomes (Kipling et al, 1994), and the centromeres of X and Y chromosomes are distal to the PAR (Fig. 1C, a′–a″, yellow arrows). In *Spdya^{cKO}* pachytene spermatocytes, the centromere of the Y chromosome, as shown by staining with the ACA antibody, was frequently located on the X axis (Fig. 1C, b′–b″, dashed circle). This indicates that the Y chromosome axis was fully aligned with the X axis. Indeed, continuous signals of SYCE1 (synaptonemal complex central element protein 1), SYCP1, and TEX12 (testis expressed 12) were observed along the entire Y axis (Fig. 1D,b–d′, yellow arrows; and arrowheads pointing at centromeres of X and Y chromosomes). This is indicative of an NH X-Y synapsis, which is a transient X-Y configuration typically observed at early pachytene (Fig. EV1A-a). We termed such an X-Y configuration in *Spdya^{cKO}* pachytene cells a "*full Y-X NH synapsis*". This is distinct from the PAR-synapsed X and Y chromosomes in control *Spdya^{fl/fl}* cells (Fig. 1D, a–a′), where the centromeres of the X and Y chromosomes remain apart from each other (Fig. 1D, a–a′, arrowheads). Quantitative analysis revealed that 47.1% of *Spdya^{cKO}* pachytene cells exhibited *full Y-X NH synapsis*, compared to only 3.3% of *Spdya^{fl/fl}* cells (Fig. 1C-c).

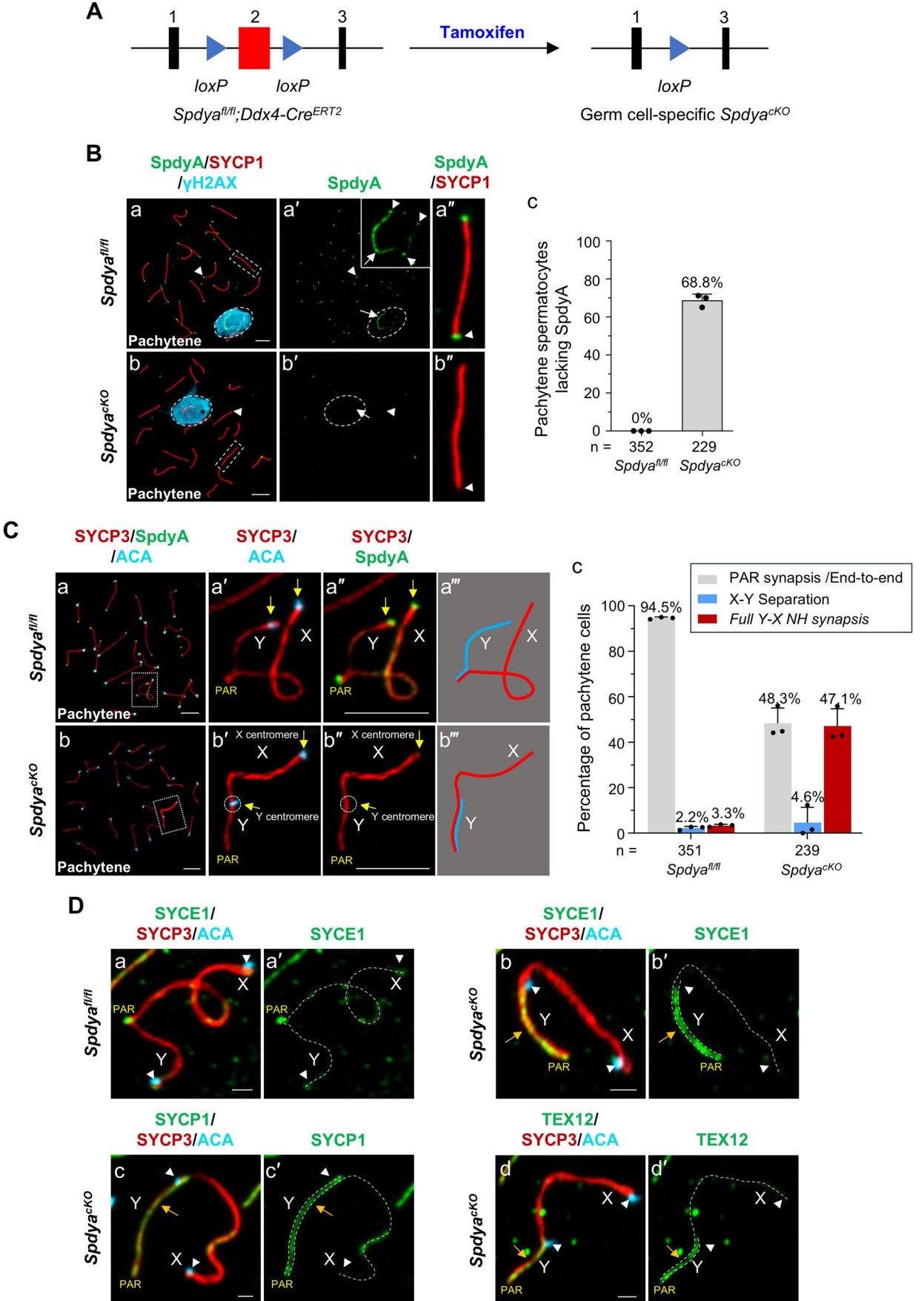

**Figure 1.  Specific *Spdya* deletion resulted in *full Y-X NH synapsis* in mouse pachytene spermatocytes.**

(A) illustration of the tamoxifen-induced deletion of exon 2 of *Spdya* in the germ cells of *Spdya*$^{fl/fl}$;*Ddx4-Cre*$^{ERT2}$ mice. (B) Chromosome spreads of *Spdya*$^{fl/fl}$ (a–a″) and *Spdya*$^{cKO}$ (b–b″) spermatocytes immunostained for SpdyA (green), SYCP1 (red) and γH2AX (light blue). Scale bars, 5 μm. (c) Percentages of pachytene spermatocytes lacking SpdyA in testes from tamoxifen-treated *Spdya*$^{fl/fl}$ and *Spdya*$^{fl/fl}$;*Ddx4-Cre*$^{ERT2}$ mice. (C) *Spdya*$^{fl/fl}$ (a–a″) and *Spdya*$^{cKO}$ (b–b″) pachytene spermatocytes immunostained for SpdyA (green), SYCP3 (red), and ACA (light blue). Compared to PAR-synapsed X-Y configuration in *Spdya*$^{fl/fl}$ cells (a, dashed rectangle is enlarged in (a, a″), illustrated in a‴), X and Y chromosomes in *Spdya*$^{cKO}$ cells exhibited *full Y-X NH synapsis* (b, dashed rectangle is enlarged in (b′–b″), illustrated b‴). Yellow arrows indicate centromeres of sex chromosomes. Scale bars, 5 μm. (c) Percentages of pachytene cells with different X-Y configurations in *Spdya*$^{fl/fl}$ and *Spdya*$^{cKO}$ testes. (D) *Spdya*$^{fl/fl}$ (a–a′) and *Spdya*$^{cKO}$ (b–d′) spermatocytes immunostained for SYCE1 (a–b′), SYCP1 (c–c′) or TEX12 (d–d′, green), SYCP3 (red) and ACA (light blue). White arrowheads indicate centromeres of X and Y chromosomes. Yellow arrows indicate continuous SYCE1 (b–b′), SYCP1 (c–c′) or TEX12 (d–d′) along the entire Y chromosome axis in *Spdya*$^{cKO}$ cells exhibiting *full Y-X NH synapsis*. Scale bars, 1 μm. Data information: In (B, C), data are presented as the mean ± SD. "n" is the total number of pachytene spermatocytes scored from three mice per genotype. Source data are available online for this figure.

## High frequency of *full Y-X NH synapsis in Spdya*$^{cKO}$ mice with synchronized meiotic prophase I

Under normal physiological conditions, the progression of meiotic prophase I in mouse spermatocytes is asynchronous. To compare cells at the same substages, we adopted a well-established mouse model with synchronized meiotic prophase I (Hogarth et al, 2013; Romer et al, 2018). This was achieved by administering a retinoic acid (RA) synthesis inhibitor to neonatal male mice, followed by RA treatment at PD9, designated as RA0 (Appendix Fig. S1A). As summarized in Appendix Fig. S1A, in the synchronization model, the male germ cells entered the pachytene stage at RA9 (i.e., PD18), which lasted about 4.5 days. The spermatocytes then entered the diplotene stage at RA13.5 (i.e., PD23) (for details, see Supplementary Result 1 and Appendix Fig. S1B).

To achieve synchronized deletion of SpdyA in mid-pachytene cells, we administered two consecutive i.p. tamoxifen injections (20 mg/kg body weight) at RA7.25 and RA8.25 (i.e., PD16 and PD17) and collected *Spdya*$^{fl/fl}$;*Ddx4-Cre*$^{ERT2}$ testicular cells at RA11.25 (i.e., PD20, Fig. 2A). Based on SYCP3 and SYCP1 staining (Fig. 2B, a–b), almost all cells in the synchronized (*Sync*)-*Spdya*$^{fl/fl}$ group and *Sync-Spdya*$^{fl/fl}$;*Ddx4-Cre*$^{ERT2}$ group were pachytene spermatocytes (Fig. 2B-c). In the *Sync-Spdya*$^{fl/fl}$;*Ddx4-Cre*$^{ERT2}$ mice, 97.2% ($n = 677$) of pachytene cells lacked SpdyA protein at telomeres (Fig. 2B-b, arrowhead) and sex chromosome axes (Fig. 2B-b, arrow, 2B-d). These cells were termed *Sync-Spdya*$^{cKO}$ pachytene cells.

We immediately noticed that at RA11.25, numerous *Sync-Spdya*$^{cKO}$ pachytene cells exhibited *full Y-X NH synapsis* (Fig. 2C), as indicated by the Y chromosome centromere ACA signal on the X axis (Fig. 2C-b, dashed circle). Quantification analysis showed that 70.1% of *Sync-Spdya*$^{cKO}$ pachytene cells exhibited *full Y-X NH synapsis*, while only 27.4% exhibited PAR synapsis (Fig. 2C-c). As a control, 96.1% of *Sync-Spdya*$^{fl/fl}$ pachytene cells displayed PAR-synapsed X-Y configurations (Fig. 2C, a, c).

## The *full Y-X NH synapsis* in *Sync-Spdya*$^{cKO}$ pachytene cells was not a result of retarded pachynema progression

We wondered whether the high frequency of *full Y-X NH synapsis* in *Sync-Spdya*$^{cKO}$ pachytene cells was caused by a delayed progression from early to mid-late pachynema. By immunostaining of ANKRD31 (ankyrin repeat domain-containing 31), we found that at RA9.25 (i.e., at early pachytene stage), similar to *Sync-Spdya*$^{fl/fl}$ pachytene cells, *Sync-Spdya*$^{cKO}$ pachytene cells with *full Y-X NH synapsis* exhibited a single ANKRD31 blob at the PAR

(Appendix Fig. S2A-b, arrow). This suggests the normal recruitment of the RMMAI proteins to the mo-2 array in the PAR, which is indicative of normal DSB formation at the PAR in *Sync-Spdya*$^{cKO}$ cells. Moreover, at RA11.25 (i.e., at mid-pachytene stage), the majority of both *Sync-Spdya*$^{fl/fl}$ and *Sync-Spdya*$^{cKO}$ spermatocytes displayed aggregated γH2AX staining in the sex chromosome area (Appendix Fig. S2B, a–b, arrowheads). Also, 94.4% of *Sync-Spdya*$^{cKO}$ and 95.5% of *Sync-Spdya*$^{fl/fl}$ pachytene spermatocytes were H1t-positive (Appendix Fig. S2B). These data suggest that the *Sync-Spdya*$^{cKO}$ spermatocytes had reached the generally regarded mid- to late pachytene stages.

We next evaluated crossover formation in *Sync-Spdya*$^{cKO}$ pachytene cells with *full Y-X NH synapsis* by staining mid-late pachytene (RA11.50) chromosome spreads for MLH1 (MutL homolog 1). Compared to *Sync-Spdya*$^{fl/fl}$ pachytene cells, *Sync-Spdya*$^{cKO}$ cells showed a modest yet significant reduction in total MLH1 focus counts (Fig. 2D, arrows in a and b, 2D-c). However, 72.6% of the *Sync-Spdya*$^{cKO}$ cells with *full Y-X NH synapsis* exhibited one MLH1 focus in the PAR of the sex chromosomes versus 60.6% in the *Sync-Spdya*$^{fl/fl}$ control cells (Fig. 2D, arrowheads in insets of a and b; and Fig. 2D-d). These results indicate that the *full Y-X NH synapsis* in *Sync-Spdya*$^{cKO}$ pachytene spermatocytes is unlikely due to retarded pachynema progression. It seems that crossover formation in the PAR was not interrupted in *Sync-Spdya*$^{cKO}$ pachytene cells, and that Y-X NH desynapsis appeared independent from meiotic recombination.

Next, to determine whether *full Y-X NH synapsis* disrupts MSCI, we first performed bulk RNA sequencing (RNA-seq) analyses with *Sync-Spdya*$^{fl/fl}$ and *Sync-Spdya*$^{cKO}$ testes at RA11.50. The transcriptional profiles of the two groups were highly similar (Appendix Fig. S3A), with no differentially expressed (DE) protein-coding genes on sex chromosomes (0 out of 484) (Fig. 2E) and only 34 DE genes on autosomes (Dataset EV1; Appendix Fig. S3B). To focus on spermatocyte transcriptomes, we further performed single-cell RNA sequencing (scRNA-seq) analyses with *Sync-Spdya*$^{fl/fl}$ and *Sync-Spdya*$^{cKO}$ testes at RA11.75 (Fig. EV2). After filtering out low-quality cells, a total of 19,708 *Sync-Spdya*$^{fl/fl}$ and 19,767 *Sync-Spdya*$^{cKO}$ testicular cells were retained for analysis. Uniform manifold approximation and projection (UMAP) analysis showed unsupervised clustering of these cells, and 3 spermatogenic and 7 somatic cell types were identified by established cell-specific markers (Fig. EV2A) (Ernst et al, 2019; Green et al, 2018). Marker gene expression differences among these main cell types were visualized with a heatmap (Fig. EV2B). Compared to *Sync-Spdya*$^{fl/fl}$ testes, *Sync-Spdya*$^{cKO}$ testes showed a lower percentage of spermatocytes and a higher percentage of spermatogonia, indicative of spermatocyte loss (Fig. EV2C). Finally, 3623 *Sync-Spdya*$^{fl/fl}$ and 2187

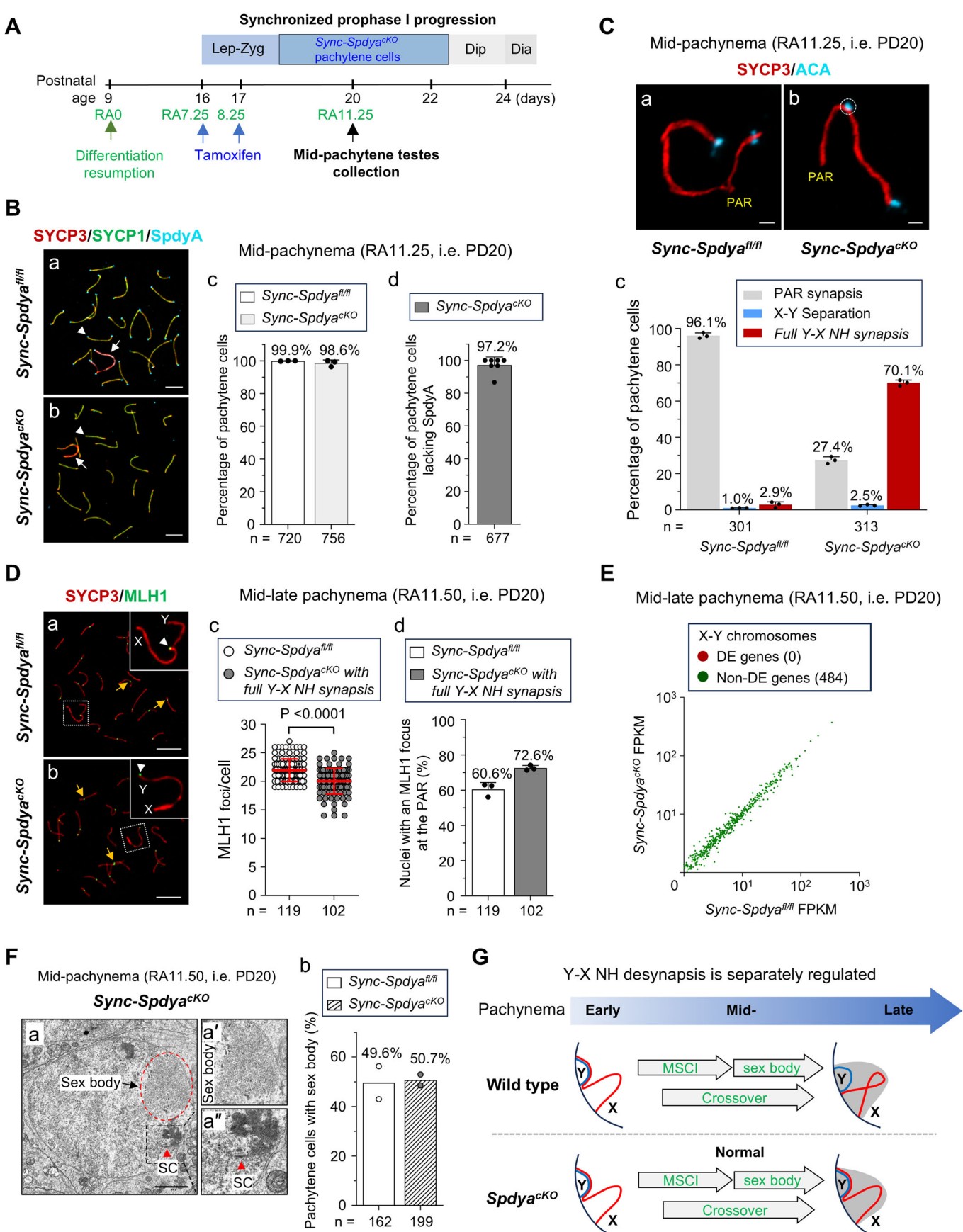

◄ **Figure 2. The *full Y-X NH synapsis* was not a result of retarded pachynema progression.**

(A) tamoxifen treatment regimen to obtain mid-pachytene *Sync-Spdya^cKO* spermatocytes in synchronized *Spdya^{fl/fl};Ddx4-Cre^{ERT2}* male mice. (B) Chromosome spreads of *Sync-Spdya^{fl/fl}* (a–a′) and *Sync-Spdya^cKO* (b–b′) spermatocytes at RA11.25, immunostained for SYCP3 (red), SYCP1 (green) and SpdyA (light blue). Arrowheads indicate telomeres and arrows indicate sex chromosome axes. Scale bars, 5 μm. (c) Percentages of pachytene cells in *Sync-Spdya^{fl/fl}* and *Sync-Spdya^cKO* testes collected at RA11.25. (d) Percentage of pachytene cells lacking SpdyA in testes from tamoxifen-treated synchronized *Spdya^{fl/fl};Ddx4-Cre^{ERT2}* mice at RA11.25. (C) Compared to PAR-synapsed X-Y in *Sync-Spdya^{fl/fl}* pachytene cells (a), sex chromosomes exhibited *full Y-X NH synapsis* in *Sync-Spdya^cKO* pachytene cells (b). Scale bars, 1 μm. (c) Percentages of pachytene cells with different X-Y configurations in *Sync-Spdya^{fl/fl}* and *Sync-Spdya^cKO* testes at RA11.25. (D) Chromosome spreads of mid-late pachytene *Sync-Spdya^{fl/fl}* (a) and *Sync-Spdya^cKO* spermatocytes (b) immunostained for SYCP3 (red) and MLH1 (green, arrows). Sex chromosomes within the dashed boxes are shown at higher magnification in the insets. Arrowheads indicate the MLH1 focus at the PAR. Scale bars, 5 μm. (c) Scatter plot showing a modest reduction in total MLH1 focus counts per cell in *Sync-Spdya^cKO* spermatocytes (closed dots) compared to *Sync-Spdya^{fl/fl}* spermatocytes (open dots). (d) Percentages of nuclei with an MLH1 focus at the PAR in *Sync-Spdya^{fl/fl}* cells and *Sync-Spdya^cKO* cells with *full Y-X NH synapsis*. (E) Transcription level comparison of sex chromosome-linked genes between *Sync-Spdya^cKO* and *Sync-Spdya^{fl/fl}* pachytene testes in bulk RNA-seq analysis. DE gene: differentially expressed gene. FPKM: Fragments Per Kilobase Million. Only protein-coding genes with an average FPKM > 1 in both groups were included in the scatter plot. (F) TEM images of representative pachytene spermatocytes in *Sync-Spdya^cKO* (a–a″) testis at RA11.50. The sex body (red dashed oval, and a′) and SC (box, and a″) were observed in *Sync-Spdya^cKO* spermatocytes. Scale bars, 2 μm. (b) Percentages of pachytene cells with sex bodies in mid-late pachytene *Sync-Spdya^{fl/fl}* and *Sync-Spdya^cKO* testes based on TEM images. (G) Illustration of the Y-X NH desynapsis as an independent process, separate from general pachynema events (i.e., MSCI, crossover, and sex body formations). Data information: In (B–D), data are presented as the mean ± SD. In (F), data are presented as the mean with individual values. "n" represents the total number of pachytene spermatocytes scored per genotype: from at least three mice in (B–D), and from two mice in (F). Statistical analyses in (D) were performed using Mann–Whitney test, and P = 0.00000000513. Source data are available online for this figure.

*Sync-Spdya^cKO* spermatocytes were subjected to DE gene analysis, and 22 autosomal genes (Dataset EV2) but no sex chromosome genes were identified as DE genes (Fig. EV2D). Therefore, both bulk RNA-seq and scRNA-seq analyses suggest that MSCI occurs normally in *Sync-Spdya^cKO* pachytene spermatocytes.

In addition, transmission electron microscopy (TEM) analyses of mid-late pachytene (RA11.50) testes revealed that sex body formation was unaffected—49.6% of observed *Sync-Spdya^{fl/fl}* pachytene cells and 50.7% of *Sync-Spdya^cKO* pachytene cells exhibited oblong sex bodies (Fig. 2F).

Collectively, these results suggest that Y-X NH desynapsis in pachytene spermatocytes is separately regulated from autosomal pachynema progression, crossover formations, MSCI, and sex body formation. Indeed, when Y-X NH desynapsis is interrupted in *Sync-Spdya^cKO* pachytene spermatocytes, crossover formation, MSCI, and sex body formation still take place normally (Fig. 2G). This suggests the presence of distinct and autonomous regulatory mechanisms for Y-X NH desynapsis.

## Continuous SpdyA function is required to maintain Tel-NE attachment and ensure spermatocyte survival throughout pachynema

We found that 47.4% of mid-pachytene *Sync-Spdya^cKO* cells (RA11.25) exhibited telomere (Tel)-NE detachment, with at least one TRF1 focus observed within the nuclear interior (Fig. EV3A-b, arrows, c). In contrast, only 11.6% of *Sync-Spdya^{fl/fl}* cells showed Tel-NE detachment (Fig. EV3A-c). Consequently, *Sync-Spdya^cKO* testes (RA13.25), in which *Spdya* was ablated for an extended period, were smaller in size and lacked late pachytene spermatocytes (Figs. EV3B, a, c, asterisk, EV3C, Regimen A). By comparison, the control *Sync-Spdya^{fl/fl}* testes contained late pachytene spermatocytes (Fig. EV3B-b). These results suggest that SpdyA is not only essential for initiating the Tel-NE tethering during zygonema, but is also critical for maintaining Tel-NE attachment and supporting spermatocyte survival throughout pachynema.

As expected, *Sync-Spdya^cKO* testes collected one month later (RA45, i.e., PD54) were smaller in size than the control *Sync-Spdya^{fl/fl}* testes and exhibited meiotic arrest at zygonema in the majority of seminiferous tubules, with a marked reduction in

tubules containing spermatids (Appendix Fig. S4A–C). Consequently, few spermatozoa were detected in the epididymides of adult *Sync-Spdya^cKO* mice (Appendix Fig. S4D), which may result in infertility.

## Non-occurrence of X-Y desynapsis in late pachytene *Sync-Spdya^cKO* spermatocytes

We next obtained late pachytene *Sync-Spdya^cKO* cells at RA13 (i.e., PD22) after a delayed tamoxifen injection (Figs. 3A and EV3C, Regimen B) and examined whether the *full Y-X NH synapsis* in *Sync-Spdya^cKO* spermatocytes affected homologous desynapsis in the PAR at late pachytene. If PAR desynapsis was unaffected, we would expect to observe separation of the PAR ends (Fig. 3B). Our results showed that compared to the normal PAR desynapsis in late pachytene *Sync-Spdya^{fl/fl}* cells which led to an end-to-end associated X-Y chromosome configuration (Fig. 3C, a–a′, red arrowhead), a proportion of late pachytene *Sync-Spdya^cKO* cells still exhibited *full Y-X NH synapsis* with continuous SYCP1 signals along the entire Y axis (Fig. 3C, b–b′, arrow). Quantification analysis revealed that 34.3% of late pachytene *Sync-Spdya^cKO* cells still exhibited *full Y-X NH synapsis* with non-occurrence of X-Y desynapsis (Fig. 3C-c).

To determine whether the *Sync-Spdya^cKO* cells at RA13 were actually arrested at the mid-late pachytene stage, we stained the chromosome spreads with histone variant macroH2A1, a marker for late pachytene(Hoyer-Fender et al, 2004; Turner et al, 2001). In wild-type *Sync-Spdya^{fl/fl}* spermatocytes, the macroH2A1 signal was very weak on the pericentric heterochromatin (PCH) of the Y chromosome at mid-late pachytene (Fig. 3D-a, arrowhead, RA11.75), but it became highly enriched in this region by late pachytene (Fig. 3D-b, dashed circle, RA13). Surprisingly, 78.2% of late pachytene *Sync-Spdya^cKO* cells with *full Y-X NH synapsis* exhibited accumulated macroH2A1 on the PCH of the Y chromosome (Fig. 3D-d, dashed circle), which was comparable to the 82.6% of *Sync-Spdya^{fl/fl}* cells showing the same pattern of macroH2A1 (Fig. 3D-e). These data indicate that the RA13 *Sync-Spdya^cKO* cells with *full Y-X NH synapsis* had indeed progressed to the late pachytene stage; however, their sex chromosomes still failed to undergo X-Y desynapsis, a process that seems to be autonomous and separately regulated.

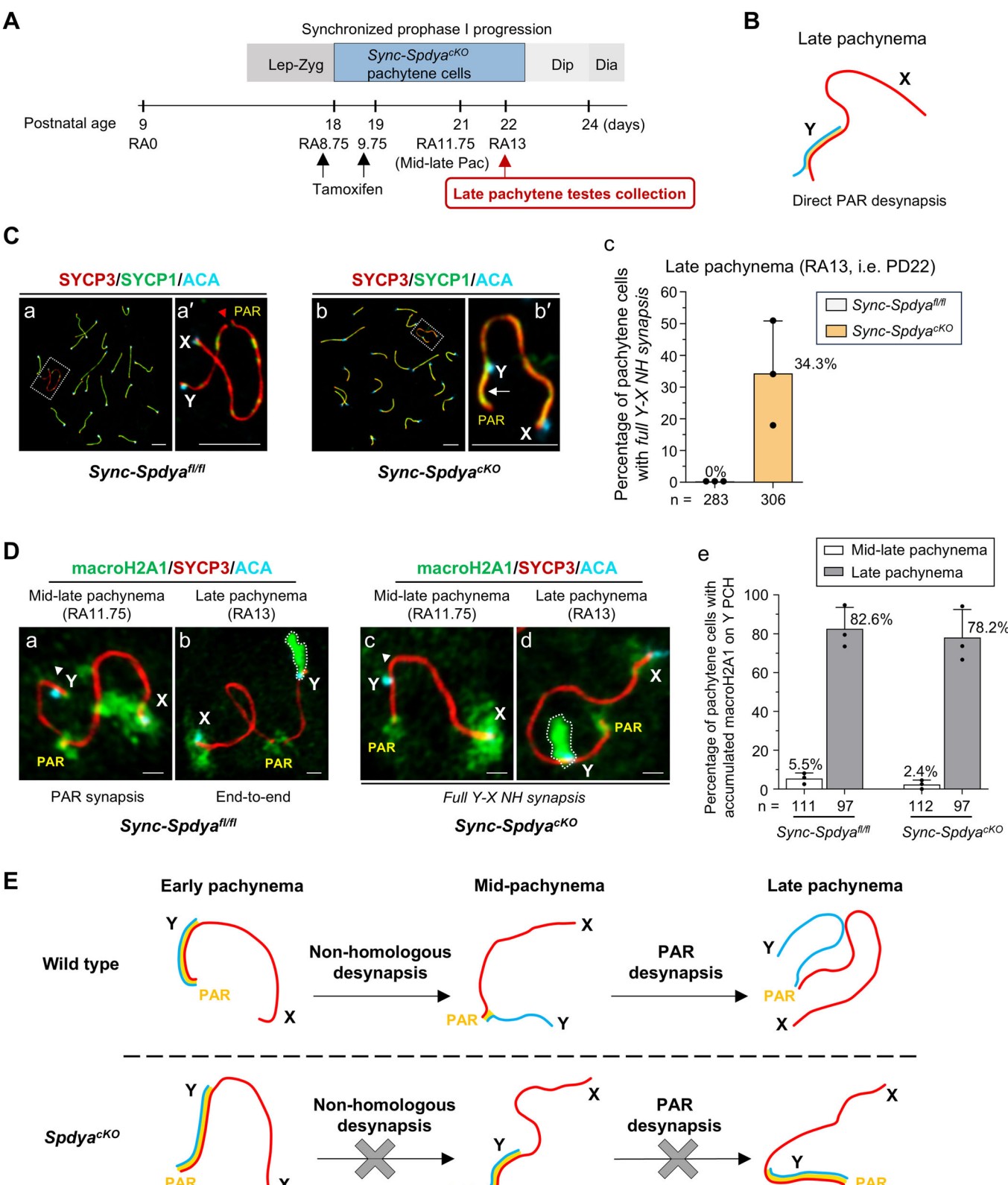

◄ **Figure 3. The persistent Y-X NH synapsis led to non-occurrence of X-Y desynapsis in late pachytene Sync-Spdya^cKO spermatocytes.**

(A) tamoxifen treatment regimen to obtain late pachytene *Sync-Spdya^cKO* spermatocytes in synchronized *Spdya^fl/fl;Ddx4-Cre^ERT2* male mice. (B) illustration of direct PAR desynapsis without Y-X NH desynapsis, a configuration ruled out by the following analysis. (C) Chromosome spreads of late pachytene (RA13) *Sync-Spdya^fl/fl* (a–a') and *Sync-Spdya^cKO* (b–b') spermatocytes immunostained for SYCP3 (red), SYCP1 (green) and ACA (light blue). Scale bars, 5 μm. (c) Percentages of pachytene cells exhibiting *full Y-X NH synapsis* in late pachytene *Sync-Spdya^fl/fl* and *Sync-Spdya^cKO* testes at RA13. (D) Chromosome spreads of mid-late and late pachytene *Sync-Spdya^fl/fl* (a, b) and *Sync-Spdya^cKO* (c, d) spermatocytes immunostained for MacroH2A1 (green), SYCP3 (red) and ACA (light blue). Arrowheads indicate the PCH region of Y chromosome at mid-late pachynema (a, c), which exhibited low abundance of MacroH2A1. However, similar to late pachytene *Sync-Spdya^fl/fl* cells (b), MacroH2A1 was accumulated on the PCH region of Y chromosome (dashed circle) in late pachytene *Sync-Spdya^cKO* cells. Scale bars, 5 μm. (e) Percentages of *Sync-Spdya^fl/fl* and *Sync-Spdya^cKO* pachytene cells with accumulated macroH2A1 on the Y PCH. (E) Illustration of the failures of Y-X NH desynapsis and X-Y desynapsis in *Spdya^cKO* pachytene spermatocytes. Data information: In (C, D), data are presented as the mean ± SD. "n" represents the total number of pachytene spermatocytes scored from three mice per genotype. Source data are available online for this figure.

Immunostaining of late pachytene *Sync-Spdya^cKO* testis sections with γH2AX and c-PARP (cleaved poly ADP ribose polymerase), an apoptosis marker, revealed a significant increase in c-PARP-positive pachytene cells (Fig. EV3D-b, arrows), indicating apoptosis of *Sync-Spdya^cKO* spermatocytes at late pachynema.

As summarized in Fig. 3E, in wild-type spermatocytes, the Y-X NH synapsis and the subsequent NH desynapsis are natural processes during early-to-mid-pachynema, which is followed by PAR desynapsis by late pachynema. In *Spdya^cKO* spermatocytes, however, the X and Y chromosomes underwent normal Y-X NH synapsis but failed to proceed with NH desynapsis, remaining fully aligned at mid-pachynema (Fig. 3E). Consequently, by late pachynema, PAR desynapsis also failed, resulting in a complete failure of X-Y desynapsis (Fig. 3E). In contrast, our results indicate that SpdyA is dispensable for autosomal desynapsis (Appendix Fig. S5).

## The persistent *full Y-X NH synapsis* interfered with sex chromosomal loop-axis organization within the sex body

To explore the functional significance of the Y-X NH desynapsis in pachytene spermatocytes, we looked into the X-Y loop-axis organization as a proxy of chromosome compaction within the sex body at pachynema. By measuring the lengths of the X and Y axes (Fig. 4A, a–b), we found that while X axis length was only modestly shortened (Fig. 4A–c), both the Y axis length and the Y/X axis ratio were significantly reduced in late pachytene *Sync-Spdya^cKO* cells with *full Y-X NH synapsis*, compared to *Sync-Spdya^fl/fl* cells (Fig. 4A, d–e). These results indicate that *full Y-X NH synapsis* may perturb the lengthening of the entire Y axis, and to a lesser extent the X axis, in *Sync-Spdya^cKO* cells during late pachynema, whereas X-Y axis lengthening proceeds normally in *Sync-Spdya^fl/fl* cells.

We hypothesized that the Y-X NH desynapsis is important for Y axis lengthening and the X-Y loop-axis organization that are needed for the sex chromosomal compaction within the pachytene sex body. As shown by immuno-FISH staining of γH2AX and Chr Y, in wild-type *Sync-Spdya^fl/fl* cells the entire Y chromosome with all its chromatin loops was enclosed within the aggregated γH2AX area (Fig. 4B, a–a''); however, in *Sync-Spdya^cKO* cells with *full Y-X NH synapsis*, part of the irregular Y loops was found outside of the γH2AX area (Fig. 4B, b–b'', dashed line-enclosed part of the Y loops). By measuring the fluorescence intensity of Y loops, we found that an average of 39.2% of the Y chromatin loop fluorescent signals in *Sync-Spdya^cKO* cells was located outside the γH2AX area (Fig. 4B-c), showing that the γH2AX phosphorylation on the Y loops was inadequate in late pachytene *Sync-Spdya^cKO* cells.

Notably, compared to the oval-shaped γH2AX domain in *Sync-Spdya^fl/fl* cells, the γH2AX domain in *Sync-Spdya^cKO* cells also appeared irregular, implying that the X chromosome compaction might also be disrupted (Fig. 4B, a–b'').

We next studied whether the disrupted Y loop-axis organization was associated with aberrant post-translational modifications within the sex body in late pachytene *Sync-Spdya^cKO* cells. The germline-specific Polycomb protein SCML2 (Sex comb on midleg-like 2) is recruited to X-Y chromatin at the early-to-mid-pachytene transition, suppressing the ubiquitination of H2A Lysine 119 (K119) on the X-Y chromosomes (Hasegawa et al, 2015; Luo et al, 2015). As shown in Fig. 4C, a–a'', the Y chromatin signals were within the SCML2 area in *Sync-Spdya^fl/fl* cells, showing that SCML2 was correctly loaded onto the Y loops. However, in late pachytene *Sync-Spdya^cKO* cells with *full Y-X NH synapsis*, the Y loops were located largely outside the SCML2 signals on sex chromosomes (Fig. 4C, b–b'', dashed line-enclosed part of the Y loops). In these *Sync-Spdya^cKO* cells, an average of 47.8% of the Y chromatin fluorescent signals per cell were SCML2-negative (Fig. 4C-c), showing that part of the Y chromatin lacked SCML2-mediated suppression of H2A K119 ubiquitination. Similar to the γH2AX staining (Fig. 4B, b–b'), the SCML2 domain also appeared dispersed and irregular in *Sync-Spdya^cKO* cells (Fig. 4C, b–b'), indicative of disrupted X loops. Similar to *Sync-Spdya^fl/fl* cells, late pachytene *Sync-Spdya^cKO* cells with end-to-end associated X-Y chromosomes showed compact Y chromatin loops enclosed within aggregated γH2AX or SCML2 areas (Appendix Fig. S6A, B).

Moreover, immunostaining with FK2 antibody in late pachytene *Sync-Spdya^cKO* cells did not detect ubiquitinated conjugates in the *full Y-X NH synapsis* region, including the *X-Y* axes and loops (Fig. 4D-b, dashed circle). Quantification showed that as many as 59.7% of the *Sync-Spdya^cKO* cells exhibited an absence of FK2 signals at the *full Y-X NH synapsis* region (Fig. 4D-c), suggesting that the defective configuration of the Y loops might result from insufficient polyubiquitination of the Y-X NH-synapsed region, or vice versa.

As summarized in Fig. 4E, in wild-type pachytene spermatocytes, the NH desynapsed X and Y chromosomes undergo widespread polyubiquitination and dramatic loop-axis organization, including axis lengthening and chromatin loop shortening, which leads to sufficient X-Y chromosome compaction within the sex body. However, in *Spdya^cKO* pachytene spermatocytes exhibiting *full Y-X NH synapsis*, it is presumed that because ubiquitination regulation is limited in the region with *full Y-X NH synapsis*, the X-Y axis lengthening is suppressed with aberrantly extended X-Y chromatin loops, leading to insufficient X-Y chromosome compaction within the sex body, which can be a reason for the observed

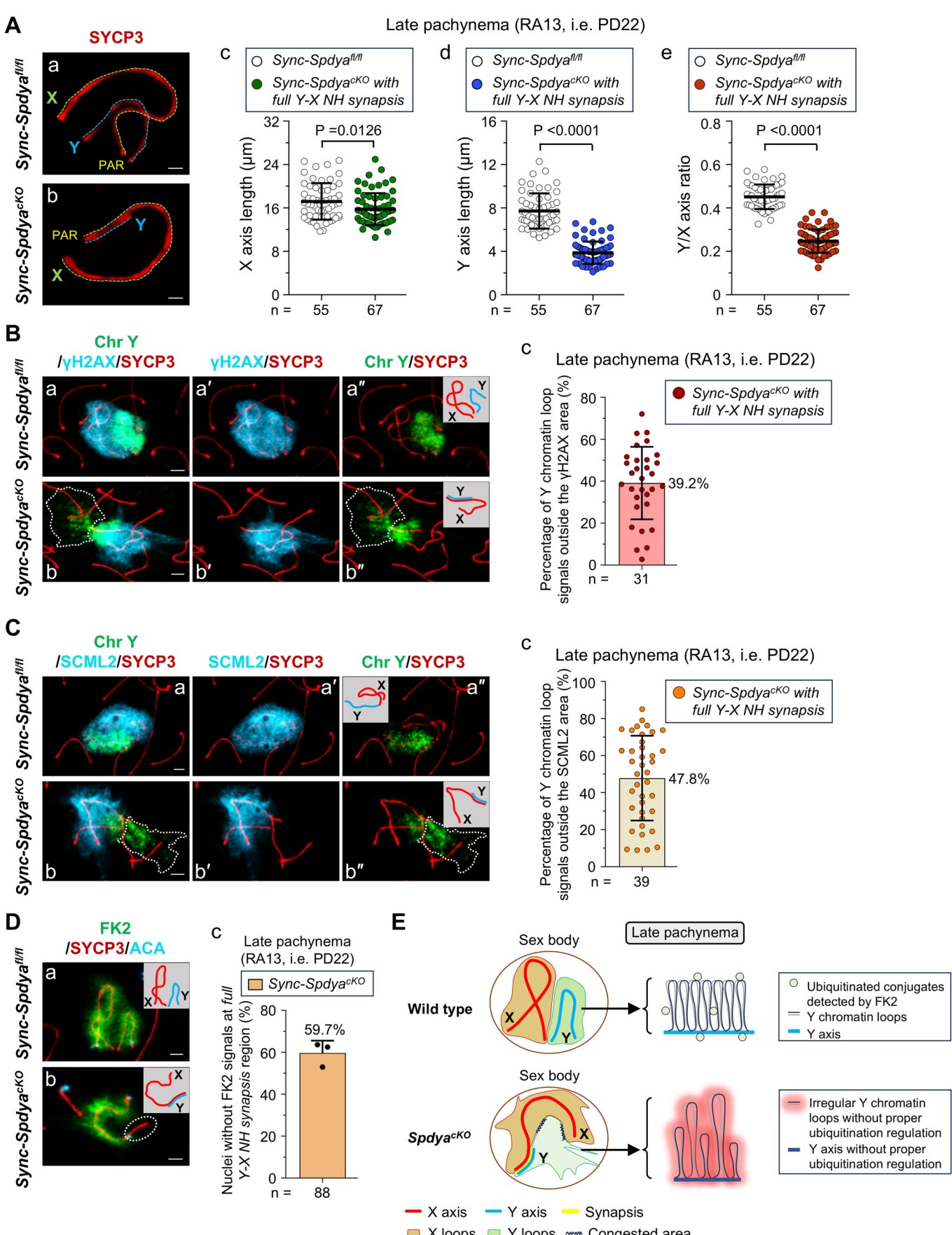

◄ **Figure 4. The persistent Y-X NH desynapsis disrupted X-Y loop-axis organization and ubiquitination within the sex body in late pachytene *Sync-Spdya^{cKO}* spermatocytes.**

(A) X-Y axis length measurements in late pachytene (RA13) *Sync-Spdya^{fl/fl}* (a) and *Sync-Spdya^{cKO}* (b) spermatocytes. Green and blue dashed lines show X and Y chromosome axes, respectively. (c–e) Measured lengths of the X axis (c), Y axis (d), and the calculated Y/X axis length ratio (e) in late pachytene *Sync-Spdya^{fl/fl}* cells and *Sync-Spdya^{cKO}* cells with *full Y-X NH synapsis*. Each dot represents a single nucleus. (B) Immuno-FISH staining of late pachytene *Sync-Spdya^{fl/fl}* (a–a″) and *Sync-Spdya^{cKO}* (b–b″) spermatocytes for γH2AX (light blue), Chr Y (green) and SYCP3 (red). The white dashed line-enclosed area indicates Y chromatin signals located outside the γH2AX area. Scale bars, 2 μm. (c) Percentage of Y chromatin loop signals outside γH2AX area in *Sync-Spdya^{cKO}* spermatocytes, based on the fluorescence intensity measurements of Y loops inside and outside the γH2AX domain. (C) Immuno-FISH staining of late pachytene *Sync-Spdya^{fl/fl}* (a–a″) and *Sync-Spdya^{cKO}* (b–b″) spermatocytes for SCML2 (light blue), Chr Y (green) and SYCP3 (red). The white dashed line-enclosed area indicates Y chromatin signals located outside the SCML2 area. Scale bars, 2 μm. (c) Percentage of Y chromatin loop signals outside SCML2 area in *Sync-Spdya^{cKO}* spermatocytes. (D) *Sync-Spdya^{fl/fl}* (a) and *Sync-Spdya^{cKO}* (b) spermatocytes immunostained for FK2 (green), SYCP3 (red) and ACA (light blue) showing the absence of FK2 signals in the *full Y-X NH synapsis* region (dashed circle in (b)) of X-Y chromosomes in *Sync-Spdya^{cKO}* spermatocytes. Scale bars, 2 μm. (c) Percentage of late pachytene *Sync-Spdya^{cKO}* spermatocytes lacking FK2 signals at the *full Y-X NH synapsis* region. (E) Illustration of a proposed mechanism underlying SpdyA-mediated X-Y loop-axis organization at late pachynema. Data information: All values are presented as the mean ± SD. "*n*" represents the total number of pachytene spermatocytes scored per genotype: from two mice in (A–C), and from three mice in (D). Statistical analyses were performed using Mann–Whitney test, and both exact *P* values in (A-d) and (A-e) < 0.000000000000001. Source data are available online for this figure.

cell death. Therefore, we propose that Y-X NH desynapsis is required for proper X-Y loop-axis organization, which permits sex chromosome compaction within the confined space of the sex body.

## The *full Y-X NH synapsis* in *Sync-Spdya^{cKO}* pachytene cells was caused by a defective LINC complex

To investigate if the *full Y-X NH synapsis* phenotype in *Sync-Spdya^{cKO}* pachytene cells was caused by presumed telomere detachment from the NE, we analyzed paraffin sections of *Sync-Spdya^{cKO}* testes at early pachynema (RA10.25), which was when 75.5% of the *Sync-Spdya^{cKO}* pachytene spermatocytes showed *full Y-X NH synapsis* (Fig. 5A). We found that autosomal Tel-NE attachment remained intact in 88.6% of early pachytene *Sync-Spdya^{cKO}* pachytene cells, similar to *Sync-Spdya^{fl/fl}* controls (Fig. 5B). The quadruple staining of testis section with HORMAD1 (HORMA Domain-Containing 1), Lamin B1, SYCP1 and ACA antibodies (Fig. 5C, a–b′) revealed that, similar to *Sync-Spdya^{fl/fl}* cells (illustrated in Fig. 5C-a″), the fully NH-synapsed X and Y chromosomes in *Sync-Spdya^{cKO}* pachytene cells exhibited intact Tel-NE attachment (illustrated in Fig. 5C-b″). It is worth noting that, under the *full Y-X NH synapsis* configuration, the middle part of the X axis was brought very close to the NE by the non-PAR telomere of the Y chromosome (Fig. 5C, b–b″, red arrows; yellow arrowheads mark the Y centromere), which might lead to excessive tension.

Further quantitative analyses revealed that X-Y chromosomes in 93.6% of the *Sync-Spdya^{fl/fl}* pachytene cells and 93.5% of the *Sync-Spdya^{cKO}* pachytene cells with *full Y-X NH synapsis* exhibited Tel-NE attachment (Fig. 5C-c). These results show that the *full Y-X NH synapsis* in *Sync-Spdya^{cKO}* pachytene cells is not a direct consequence of Tel-NE detachment.

Next, we examined whether the persistent *full Y-X NH synapsis* in *Sync-Spdya^{cKO}* cells could be a result of weakened Tel-NE attachment. Staining of chromosome spreads collected at RA10.25 showed that the telomeric localization of TRF1, TERB1 and MAJIN were intact in *Sync-Spdya^{cKO}* pachytene cells (Appendix Fig. S7). However, compared to *Sync-Spdya^{fl/fl}* pachytene cells where SUN1 signals were observed at telomeres (Fig. 5D, a–a′, arrows), *Sync-Spdya^{cKO}* pachytene cells completely lost their SUN1 signals at telomeres of both autosomes and sex chromosomes (Fig. 5D, b–b′, arrows). These cells also lacked KASH5 signals at telomeres of both

autosomes and sex chromosomes (Fig. 5E, b–b′, arrows). These results indicate that the LINC complex became defective when the telomeres lacked SpdyA, even though Tel-NE attachment was temporarily maintained. It is possible that the persistent *full Y-X NH synapsis* in *Sync-Spdya^{cKO}* cells was caused by defective LINC complexes with weakened pulling force on the non-PAR telomeres of Y chromosome, which failed to efficiently pull the telomeres away from the X axis (Fig. 5C-b″, red arrow) for the initiation of Y-X NH desynapsis.

## Disruption of the TRF1-SpdyA-LINC axis resulted in *full Y-X NH synapsis*

To confirm that the *full Y-X NH synapsis* in *Spdya^{cKO}* pachytene spermatocytes was a result of the weakened telomere-LINC complex, we disrupted the telomere shelterin complex by deleting the *Trf1* gene in pachytene spermatocytes, also using the synchronized mouse model (Fig. 6A, for details see Supplementary Results 2 and 3 and Appendix Fig. S8). As shown in Fig. 6B, we immediately noticed a high percentage of persistent *full Y-X NH synapsis* (Fig. 6B, b–c) in 66.2% of the *Sync-Trf1^{cKO}* pachytene cells (Fig. 6B-d).

Interestingly, in *Sync-Trf1^{cKO}* pachytene cells with *full Y-X NH synapsis*, we observed three distinct SpdyA expression patterns at telomeres of sex chromosomes:

(1) The SpdyA signal was lost only from the non-PAR telomeres of the Y chromosome (Fig. 6C, a–a′, red arrows), while it remained intact at the PAR-end telomeres (Fig. 6C, a–a′, yellow arrowheads) and at the non-PAR telomeres of X chromosome (Fig. 6C, a–a′, white arrowheads).

(2) The SpdyA signal was lost from the non-PAR telomeres of both X and Y chromosomes (Fig. 6C, b–b′, red arrows), while it remained intact at the PAR-end telomeres (Fig. 6C, b–b′, yellow arrowheads).

(3) The SpdyA signal was lost from all of the telomeres of X and Y chromosomes (Fig. 6C, c–c′, white arrows).

Quantitative analysis revealed that among the *Sync-Trf1^{cKO}* pachytene cells with *full Y-X NH synapsis*, 42.0% lacked SpdyA specifically at the non-PAR telomeres of the Y chromosome, 51.7% lacked SpdyA at the non-PAR telomeres of both X and Y chromosomes, and 6.3% lacked SpdyA at all telomeres (Fig. 6C, D).

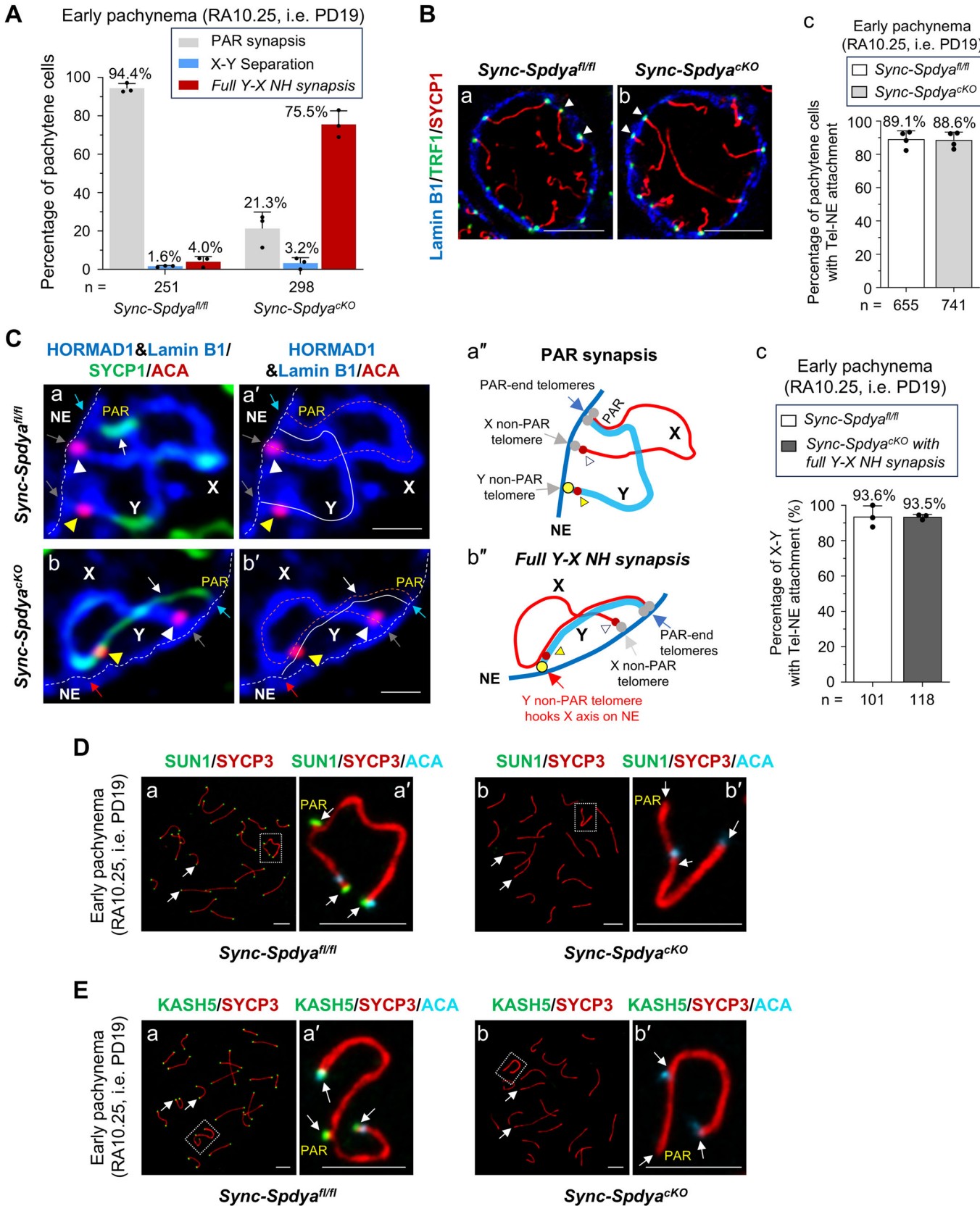

**Figure 5. The *full Y-X NH synapsis* in *Sync-Spdya^{cKO}* pachytene cells was caused by defective LINC complex.**

(A) Percentages of pachytene cells with different X-Y configurations in early pachytene *Sync-Spdya^{fl/fl}* and *Sync-Spdya^{cKO}* testes collected at RA10.25, following the tamoxifen regimen shown in Fig. 2A. (B) Early pachytene (RA10.25) *Sync-Spdya^{fl/fl}* (a) and *Sync-Spdya^{cKO}* (b) testis section immunostained for TRF1 (green), SYCP1 (red) and Lamin B1 (blue), showing intact autosomal Tel-NE attachment (white arrowheads) in *Sync-Spdya^{cKO}* spermatocytes at early pachynema. Scale bars, 5 μm. (c) Percentages of pachytene cells with Tel-NE attachment in *Sync-Spdya^{fl/fl}* and *Sync-Spdya^{cKO}* testes at RA10.25. (C) IF analyses of RA10.25 *Sync-Spdya^{fl/fl}* (a, a') and *Sync-Spdya^{cKO}* (b, b') testis sections with Lamin B1 (blue, for labeling the NE as indicated by the white dashed line), HORMAD1 (also blue, for labeling the unsynapsed region of the X and Y chromosome axes), SYCP1 (green, for labeling the synapsed region as indicated by the white arrows), and ACA (red, for labeling the centromeres of X and Y chromosomes, as indicated by the white and yellow arrowheads, respectively). (a', b') The orange dashed lines and solid white lines indicate the X and Y chromosome axes, respectively. Scale bars, 1 μm. (a″, b″) Illustration of Tel-NE attachment of X-Y chromosomes in early pachytene *Sync-Spdya^{fl/fl}* (a″) and *Sync-Spdya^{cKO}* (b″) spermatocytes. (c) Comparable percentages of X-Y chromosomes with Tel-NE attachment in *Sync-Spdya^{fl/fl}* and *Sync-Spdya^{cKO}* pachytene spermatocytes. (D) *Sync-Spdya^{fl/fl}* (a-a') and *Sync-Spdya^{cKO}* (b-b') pachytene spermatocytes immunostained for SUN1 (green), SYCP3 (red) and ACA (light blue). Scale bars, 5 μm. (E) *Sync-Spdya^{fl/fl}* (a-a') and *Sync-Spdya^{cKO}* (b-b') pachytene spermatocytes immunostained for KASH5 (green), SYCP3 (red) and ACA (light blue). Scale bars, 5 μm. Data information: All values are presented as the mean ± SD. "*n*" represents the total number of pachytene spermatocytes scored from three mice per genotype. Source data are available online for this figure.

Interestingly, in *Sync-Trf1^{cKO}* pachytene cells with *full Y-X NH synapsis,* we also observed three SUN1 expression patterns on their sex chromosomes:

(1) The SUN1 signal was lost from the non-PAR telomeres of Y chromosomes (Fig. 6D, a–a', red arrows), whereas it was still present at the PAR-end telomeres (Fig. 6D, a–a', yellow arrowheads) and at the non-PAR telomeres of the X chromosomes (Fig. 6D, a–a', white arrowheads).

(2) The SUN1 signal was lost from the non-PAR telomeres of both X and Y chromosomes (Fig. 6D, b–b', red arrows), whereas it was still present at the PAR-end telomeres (Fig. 6D, b–b', yellow arrowheads).

(3) The SUN1 signal was lost from all telomeres (Fig. 6D, c–c', white arrows).

Quantification showed that 49.1% of the *Sync-Trf1^{cKO}* pachytene cells with *full Y-X NH synapsis* lacked SUN1 at non-PAR telomeres of the Y chromosome, 39.7% lacked SUN1 at non-PAR telomeres of both X and Y chromosomes, and 11.2% lacked SUN1 at all telomeres (Fig. 6D-d).

These data suggest that the non-PAR telomeres of the X and Y chromosomes are more vulnerable to TRF1 disruption than the PAR-end telomeres. The loss of TRF1 leads to the loss of SpdyA and SUN1, which occurred more often at the non-PAR telomeres, especially from the non-PAR telomeres of the Y chromosome. This is likely the reason for the impaired Y-X NH desynapsis in *Sync-Trf1^{cKO}* pachytene cells.

## Pachytene spermatocytes were vulnerable to hypomorphic-SpdyA-induced *full Y-X NH synapsis*

We generated a hypomorphic-SpdyA mouse model, the *Spdya^{A125V}* knock-in mice, which recapitulate a missense mutation originally identified in a patient with non-obstructive azoospermia (NOA, Appendix Fig. S9). The SpdyA^{A125V} protein was unstable and had a shortened half-life (see Supplementary Result 4 and Appendix Fig. S10). Adult *Spdya^{A125V}* male mice exhibited reduced testis size and a marked decrease in elongated spermatids compared to their *Spdya^{+/+}* littermates (Fig. EV4A, B). In *Spdya^{A125V}* pachytene cells, SpdyA signals were reduced at telomeres and along the sex chromosome axes (Fig. EV4C), and up to 47.0% of these cells exhibited *full Y-X NH synapsis*, as well as 11.5% X-Y separation (Fig. EV4D). Quantitative analysis of meiotic prophase I stages

revealed a significant reduction in diplotene spermatocytes in *Spdya^{A125V}* testes (Fig. EV4E), likely due to pachytene spermatocyte apoptosis observed in *Spdya^{A125V}* seminiferous tubules (Fig. EV4F).

We further found that *Spdya^{A125V}* pachytene spermatocytes exhibited normal Tel-NE attachment (Fig. EV4G), as well as normal H1t replacement and crossover formation (Appendix Fig. S11). However, in late pachytene *Spdya^{A125V}* cells with *full Y-X NH synapsis*, the loop-axis organization of the Y chromosome —and possibly also the X chromosome—was disrupted (Fig. EV4H). This defect was not observed in *Spdya^{A125V}* cells with end-to-end associated sex chromosomes (Appendix Fig. S6C). Based on these findings, we conclude that the disrupted loop-axis organization of sex chromosomes, caused by persistent *full Y-X NH synapsis*, is likely one of the key contributors to the elimination of *Spdya^{A125V}* pachytene cells, and that pachytene spermatocytes are particularly vulnerable to hypomorphic-SpdyA function.

## Phosphoproteomic screening revealed the phosphorylation of Serine 48 of SUN1 as a key function of SpdyA in mediating Y-X NH desynapsis

SpdyA is characterized as an activator of CDK1/2 (Gonzalez and Nebreda, 2020). To identify the physiological substrates of SpdyA/CDK2 kinases in mediating Y-X NH desynapsis, we performed phosphoproteomic analyses on testes from *Sync-Spdya^{fl/fl}* and *Sync-Spdya^{cKO}* mice, collected at RA10.25, when over 75% of *Sync-Spdya^{cKO}* pachytene cells exhibit *full Y-X NH synapsis* (Figs. 7A and 5A). The phosphoproteomic screening identified five telomere-associated proteins with altered phosphorylation levels, all of which were down-regulated in *Sync-Spdya^{cKO}* testes (Fig. 7B-a). Among them, the phosphorylation of SUN1 at Serine 48 [p-SUN1(S48)] was of particular interest because SUN1 is a SpdyA-binding protein and is a critical component of the LINC complex (Fig. 7B-b) (Chen et al, 2021). Phosphoproteomic analysis revealed a 70.8% reduction in p-SUN1(S48) levels in *Sync-Spdya^{cKO}* cells, with unchanged total SUN1 protein levels as measured by a simultaneous proteomic analysis (Fig. 7C).

To detect the levels of p-SUN1(S48), we first generated a rabbit polyclonal antibody against this phosphorylation site using a synthetic phosphorylated peptide LEPVFD (pS) PRMSR as the antigen. The specificity of the antibody was validated using multiple approaches. First, the antibody detected a p-SUN1(S48) band by Western blot in HEK293T cells transfected with *pcDNA3.1-Cdk2-Myc* and *pcDNA3.1-Spdya^{WT}-HA* plasmids together with only

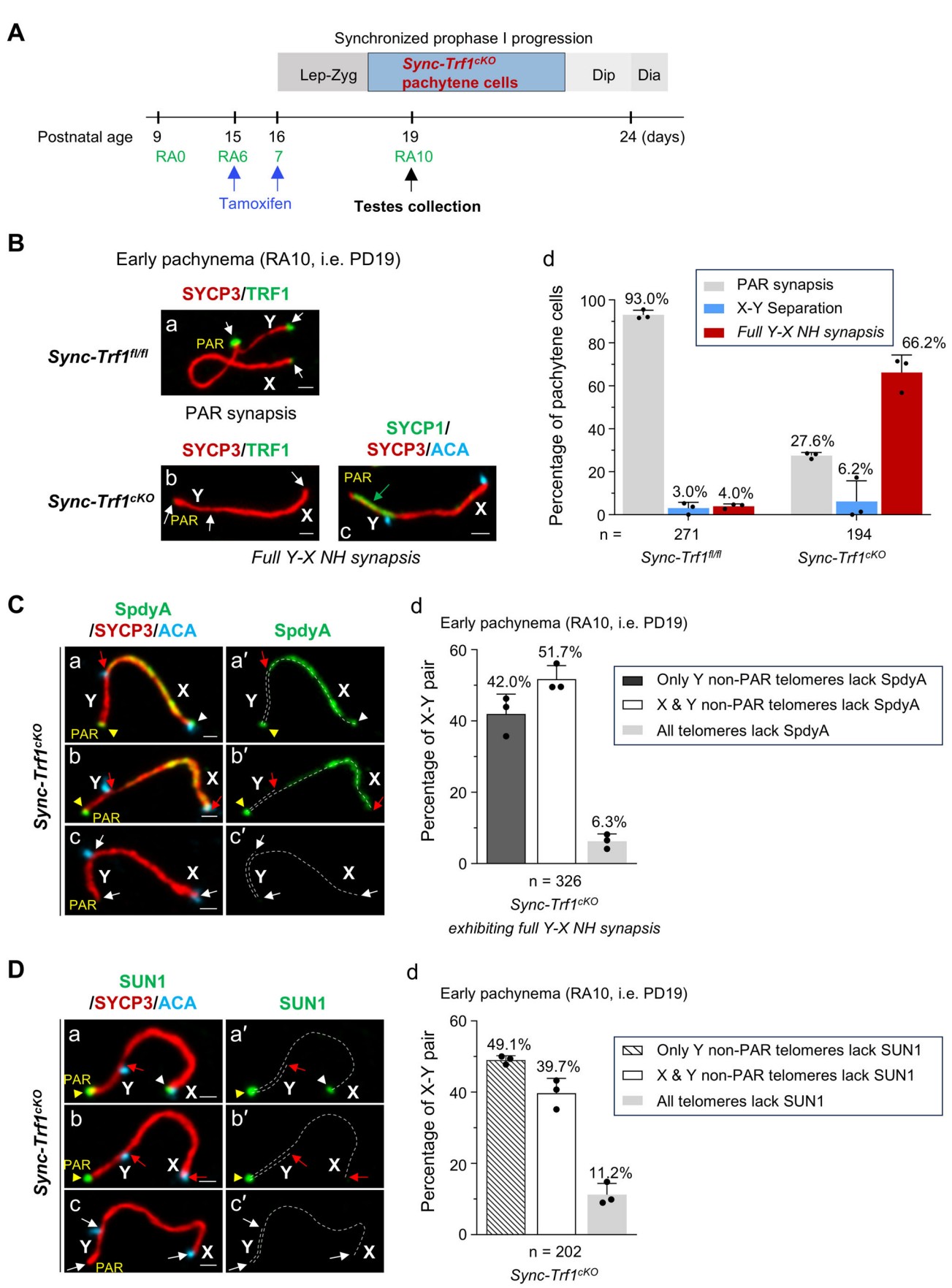

**A** Synchronized prophase I progression

**B** Early pachynema (RA10, i.e. PD19)

*Sync-Trf1^{fl/fl}*

SYCP3/TRF1

a — PAR synapsis

*Sync-Trf1^{cKO}*

SYCP3/TRF1 — b

SYCP1/SYCP3/ACA — c

Full Y-X NH synapsis

**d**

**C** SpdyA/SYCP3/ACA — SpdyA

*Sync-Trf1^{cKO}*

**d** Early pachynema (RA10, i.e. PD19)

Only Y non-PAR telomeres lack SpdyA — 42.0%
X & Y non-PAR telomeres lack SpdyA — 51.7%
All telomeres lack SpdyA — 6.3%

n = 326
*Sync-Trf1^{cKO}*
exhibiting full Y-X NH synapsis

**D** SUN1/SYCP3/ACA — SUN1

*Sync-Trf1^{cKO}*

**d** Early pachynema (RA10, i.e. PD19)

Only Y non-PAR telomeres lack SUN1 — 49.1%
X & Y non-PAR telomeres lack SUN1 — 39.7%
All telomeres lack SUN1 — 11.2%

n = 202
*Sync-Trf1^{cKO}*
exhibiting full Y-X NH synapsis

◄

**Figure 6. Pachytene stage-specific disruption of TRF1 resulted in *full Y-X NH synapsis*.**

(A) tamoxifen treatment regimen to obtain *Sync-Trf1$^{cKO}$* pachytene spermatocytes in synchronized *Trf1$^{fl/-}$;Ddx4-Cre$^{ERT2}$* male mice. (B) Compared to *Sync-Trf1$^{fl/fl}$* pachytene cells (a) that exhibited PAR synapsis between X and Y chromosomes, large numbers of *Sync-Trf1$^{cKO}$* pachytene cells (b) exhibited *full Y-X NH synapsis*, as confirmed by continuous SYCP1 signals along the entire Y chromosome axis (c, green arrow). White arrow indicates the telomere. (d) Percentages of pachytene cells with different X-Y configurations in *Sync-Trf1$^{fl/fl}$* and *Sync-Trf1$^{cKO}$* testes at RA10. (C) In *Sync-Trf1$^{cKO}$* pachytene cells with *full Y-X NH synapsis*, SpdyA (green) was frequently lost from non-PAR telomere of Y (a–a', red arrows) or both non-PAR telomeres of X and Y chromosomes (b–b', red arrows). Yellow and white arrowheads indicate the unaffected PAR-end telomeres (a–b') and non-PAR X telomeres (a–a'), respectively. A minority of sex chromosomes with *full Y-X NH synapsis* showed a loss of SpdyA from all telomeres (c–c', white arrows), as well as from the sex chromosome axes. Scale bars, 1 μm. (d) Percentages of X-Y pairs with SpdyA signal lost from different telomeres in *Sync-Trf1$^{cKO}$* pachytene cells with *full Y-X NH synapsis*. (D) In *Sync-Trf1$^{cKO}$* pachytene cells with *full Y-X NH synapsis*, SUN1 was also frequently lost from non-PAR telomeres of Y (a–a', red arrows) or both non-PAR telomeres of X and Y chromosomes (b–b', red arrows). Arrowheads indicate SUN1 signals at unaffected telomeres. A minority of sex chromosomes with *full Y-X NH synapsis* showed a loss of SUN1 from all telomeres (c–c', white arrows). Scale bars, 1 μm. (d) Percentages of X-Y pairs with SUN1 signal lost from different telomeres in *Sync-Trf1$^{cKO}$* pachytene cells with *full Y-X NH synapsis*. Data information: All values are presented as the mean ± SD. "*n*" represents the total number of pachytene spermatocytes scored from three mice per genotype. Source data are available online for this figure.

*pcDNA3.1-Sun1$^{WT}$-FLAG* plasmid (Fig. 7D), but not in cells with the *pcDNA3.1-Sun1$^{S48A}$-FLAG* plasmid or in cells lacking *pcDNA3.1-SpdyA$^{WT}$-HA* plasmid (Fig. 7D). This indicated that the phosphorylation of S48 of SUN1 was achieved by SpdyA. The specificity of this antibody was further validated by three other means as summarized in Appendix Fig. S12.

Using this specific antibody, we observed p-SUN1(S48) signals at telomeres of autosomes (Fig. 7E, a–a', arrowheads) and sex chromosomes (Fig. 7E-a'', arrows) in *Spdya$^{fl/fl}$* pachytene cells. The p-SUN1(S48) signals observed in the sex body region were non-specific, as the unphosphorylated SUN1 was exclusively located at telomeres (Fig. 7E; Appendix Fig. S12B). In *Spdya$^{cKO}$* pachytene cells, however, the p-SUN1(S48) signal was absent at telomeres of autosomes (Fig. 7E, b–b', arrowheads) and sex chromosomes (Fig. 7E-b'', arrows), consistent with our result from the phosphoproteomic analysis (Fig. 7C).

We next tested whether immunoprecipitated endogenous SpdyA from WT mouse testis could phosphorylate the SUN1 protein at S48. We immunoprecipitated HA-SpdyA from testis lysates from male mice carrying endogenous HA-SpdyA (i.e., the *Spdya$^{HA/HA}$* mice). As expected, endogenous CDK2 was also pulled down along with HA-SpdyA (Fig. 7F). The immunoprecipitates were then used for an in vitro kinase assay with recombinant 6×His-SUN1$_{1-217}$ (WT), or 6×His-SUN1$_{1-217}$ (S48A) proteins as substrates. As shown in Fig. 7F, the p-SUN1(S48) signal was only detected in the group of immunoprecipitated HA-SpdyA with 6×His-SUN1$_{1-217}$ (WT). These results confirmed that the SpdyA/CDK2 complex is able to phosphorylate SUN1 at S48.

In this context, as illustrated in Fig. 7G, we propose that the phosphorylation of SUN1 (S48) is maintained by SpdyA/CDK2 and that this is important for maintaining the integrity of the LINC complex, which in turn is required to facilitate the movement of the non-PAR telomeres of the Y chromosome in order to guide the NH desynapsis of the Y axis from the X axis (Fig. 7G). Taken together, our results suggest that the SpdyA/CDK2-mediated phosphorylation of SUN1 (S48) stabilizes the LINC complex and is thus key for driving the cytoskeleton-LINC-telomere co-movements of sex chromosomes and their NH desynapsis.

## Discussion

We and others previously reported that SpdyA plays an essential role in tethering telomeres to the NE during leptonema to zygonema in order to ensure proper autosomal synapsis and prophase I progression (Mikolcevic et al, 2016; Tu et al, 2017). In this study, we extended our focus to pachytene spermatocytes and generated a mouse model in which SpdyA was specifically knocked out in pachytene spermatocytes. We revealed a novel function of SpdyA in regulating Y-X NH desynapsis, which is surprisingly independent from the generally regarded events of pachynema progression, such as MSCI, crossover formation, and sex body formation. In addition, we found that Y-X NH desynapsis is critical for proper X-Y loop-axis organization and ubiquitination within the sex body. Therefore, the NH desynapsis of sex chromosomes is a distinct, independently regulated process that contributes to proper pachytene X-Y chromosome compaction during meiosis.

Why X and Y chromosomes undergo NH synapsis is unclear. It has been shown that partial homologous synapsis can cause NH synapsis between X[16] and 16 chromosomes in T(X;16)16H male mice (Turner et al, 2006). It is possible that PAR synapsis drives extensive Y-X NH synapsis as an automatic process without specific regulation, as indicated by the nearly complete NH synapsis between Y chromosome and the inverted X chromosome in In(X)1H male mice (Ashley, 1987). Conversely, our study shows that Y-X NH desynapsis, which promptly follows Y-X NH synapsis, is predominantly regulated by SpdyA. This implies that Y-X NH synapsis may impede some key meiotic processes, which might be resolved by Y-X NH desynapsis. A reported study using immuno-FISH staining indicated that the Y chromatin loops undergo notable compaction around the Y axis from late zygonema to pachynema (Kauppi et al, 2011). In this study, our analyses of *Spdya$^{cKO}$* spermatocytes revealed that persistent *full Y-X NH synapsis* interrupts Y axis lengthening and disrupts chromatin loop organization, leading to defective X-Y chromosome compaction within the sex body. Thus, we propose that Y-X NH desynapsis acts as an initial step in large-scale loop-axis reorganization during the pachytene stage, including axis lengthening and chromatin loop shortening as well as widespread ubiquitination on X-Y axes and loops. Therefore, this SpdyA-initiated process appears to be essential for establishing a normal X-Y configuration in pachytene spermatocytes.

One intriguing finding in our study is that the fully NH-synapsed X-Y chromosomes do not escape MSCI. Why is this the case? X-Y synapsis is highly regulated and occurs specifically at early pachynema, a stage when MSCI has already been initiated across the X-Y chromosomes. MSCI is typically established by mid-pachynema and is known to remain remarkably stable once established (Royo et al, 2013). It is possible that from early to mid-pachynema, the loop-axis reorganization of sex chromosomes is

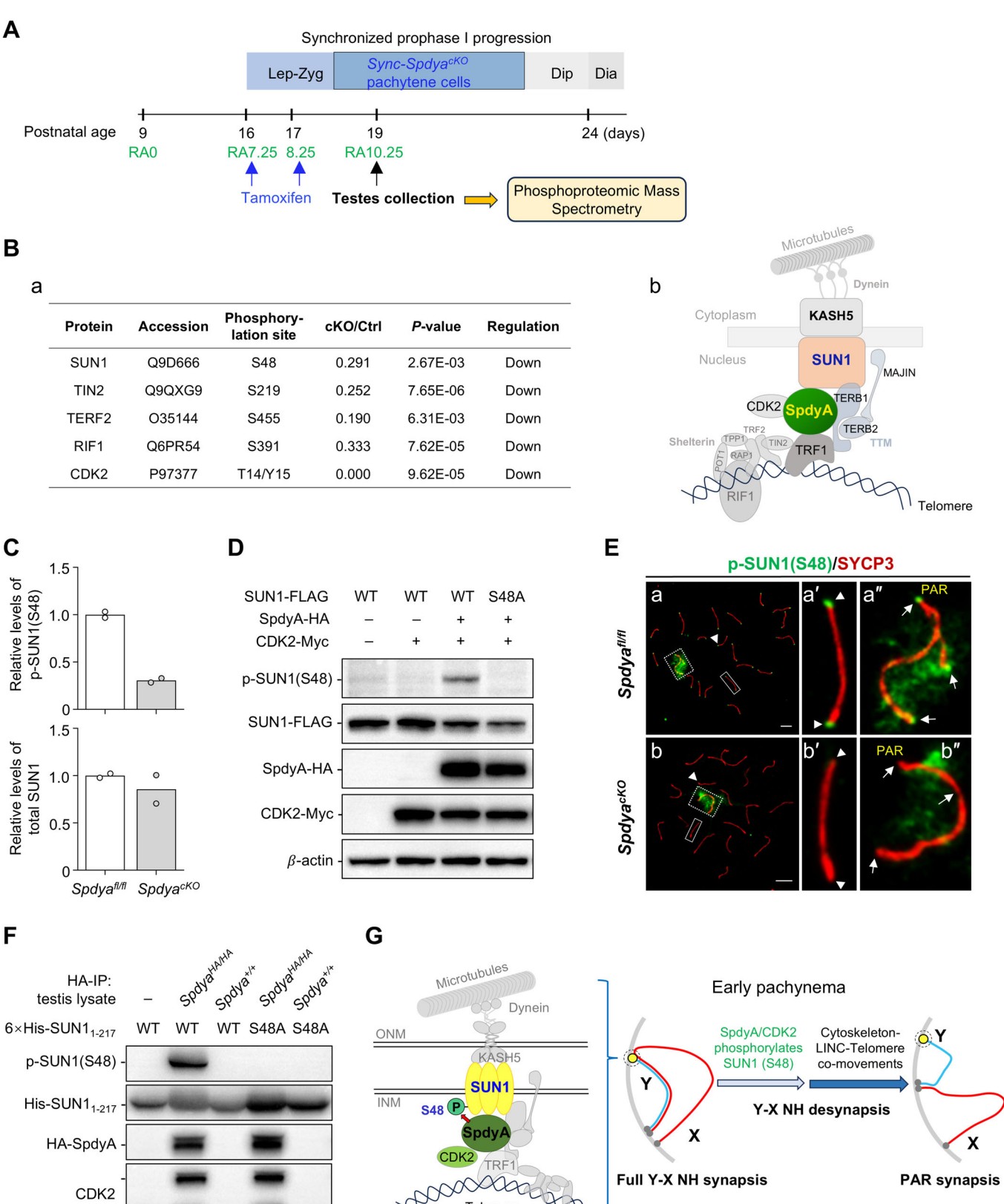

**A** Synchronized prophase I progression

**B** a

| Protein | Accession | Phosphorylation site | cKO/Ctrl | P-value | Regulation |
|---------|-----------|---------------------|----------|---------|------------|
| SUN1 | Q9D666 | S48 | 0.291 | 2.67E-03 | Down |
| TIN2 | Q9QXG9 | S219 | 0.252 | 7.65E-06 | Down |
| TERF2 | O35144 | S455 | 0.190 | 6.31E-03 | Down |
| RIF1 | Q6PR54 | S391 | 0.333 | 7.62E-05 | Down |
| CDK2 | P97377 | T14/Y15 | 0.000 | 9.62E-05 | Down |

**Figure 7. SpdyA mediated the phosphorylation of SUN1 at S48 for Y-X NH desynapsis.**

(A) The collection of synchronized testis samples at RA10.25 for phosphoproteomic mass spectrometry. (B) Five telomere-associated proteins with altered phosphorylation levels were identified (a), all of which were down-regulated in *Sync-Spdya^{cKO}* testes. (b) SpdyA-SUN1 binding is required for Tel-NE attachment. (C) Phosphoproteomic analysis revealed sharply reduced relative levels of phosphorylated S48 of SUN1 [p-SUN1(S48)] with unchanged total SUN1 protein levels in *Sync-Spdya^{cKO}* testes compared to those in *Sync-Spdya^{fl/fl}* testes at RA10.25. (D) The immunoblotting showing that the polyclonal antibody against p-SUN1(S48) detected a p-SUN1(S48) band in HEK293T cell lysates transfected with the *pcDNA3.1-Sun1^{WT}-FLAG* plasmid, but not with the *pcDNA3.1-Sun1^{S48A}-FLAG* plasmid. The p-SUN1(S48) signal was SpdyA-dependent. (E) Compared to *Spdya^{fl/fl}* cells (a-a″), p-SUN1(S48) signals were absent at the telomeres of both autosomes (b-b′, arrowheads) and sex chromosomes (b″, arrows) in *Spdya^{cKO}* pachytene cells. Scale bars, 5 μm. (F) Kinase assay showing that endogenous SpdyA/CDK2 complex, immunoprecipitated from *Spdya^{HA/HA}* testes, mediates SUN1 phosphorylation at S48. The experiment was performed with three biological replicates, and one representative result is shown. (G) Illustration of the phosphorylation of SUN1(S48) mediated by SpdyA/CDK2 at the non-PAR telomeres of Y chromosome (yellow dot in the dashed circle), which may be a key in conveying the cytoskeletal forces needed to drive the telomere movements of X-Y chromosomes and thus ensure Y-X NH desynapsis. Source data are available online for this figure.

not as pronounced as it is during the late pachytene stage, thereby allowing MSCI to be properly established in *Spdya^{cKO}* pachytene spermatocytes. This hypothesis remains to be tested in future studies.

An important unresolved question in our study is the mechanism underlying the death of spermatocytes with persistent *full Y-X NH synapsis*. At present, it remains challenging to establish a direct causal link between disrupted X-Y loop-axis organization and the cell death seen in the *Spdya^{cKO}* mouse model, which exhibits severe Tel-NE detachment that may compromise the nuclear lamina and affect spermatocyte survival (Zetka et al, 2020). However, our analysis of *Spdya^{A125V}* mice—where Tel-NE attachment, H1t replacement, and crossover formation are preserved—revealed a strong correlation between *full Y-X NH synapsis* and disrupted X-Y loop-axis organization in late pachytene spermatocytes. We also observed that a large proportion of *Spdya^{A125V}* pachytene spermatocytes failed to progress to diplonema and instead underwent apoptosis. These findings suggest a potential role for chromosomal compaction dynamics in meiotic checkpoint surveillance. We propose that further investigation using the *Spdya^{A125V}* model will help elucidate how impaired X-Y chromosomal compaction contributes to pachytene spermatocyte apoptosis.

SpdyA facilitates the anchoring of telomeres to the NE through binding with TRF1 (Mikolcevic et al, 2016; Tu et al, 2017; Wang et al, 2018). Our analyses of *Trf1^{cKO}* pachytene spermatocytes also revealed persistent *full Y-X NH synapsis*, and notably, the majority of *full Y-X NH synapsis* in *Trf1^{cKO}* pachytene cells was accompanied by the absence of SpdyA and SUN1 at the non-PAR telomeres of Y chromosomes, whereas SpdyA and SUN1 were mostly retained at the PAR-end telomeres of X-Y chromosomes. These findings underscore the critical role of intact telomere structures, especially those of the non-PAR telomeres of the sex chromosomes, as a linkage with the LINC complex in mediating Y-X NH desynapsis during early pachynema. We propose that in the configuration of *full Y-X NH synapsis*, the "non-PAR telomeres-LINC" structure of the Y chromosome experiences excessive tension because it must bear the additional tension of anchoring the paired X axis to the NE (Fig. 5C). Thus, the Y chromosomal non-PAR telomeres are more susceptible to telomere defects, which can weaken their functional connections to cytoskeletal dynamics, ultimately leading to the persistent *full Y-X NH synapsis* configuration and the non-occurrence of Y-X NH desynapsis.

In prophase I of mouse spermatocytes, the LINC complex acts as a bridge connecting telomeres to the cytoskeleton in order to regulate telomere movement along the NE (Burke, 2018; Ding et al,

2007; Horn et al, 2013; Morimoto et al, 2012). A recent study reported that SUN1-SpdyA binding is key for linking telomeres to the LINC complex for proper autosomal synapsis in prophase I spermatocytes (Chen et al, 2021). In the current work, we demonstrated that the phosphorylation of SUN1 at S48 was reduced in *Spdya^{cKO}* pachytene spermatocytes, and that mouse SpdyA/CDK2 could phosphorylate SUN1 at S48 in an in vitro kinase assay. We thus propose that this SUN1 (S48) phosphorylation is the key step in stabilizing SUN1 in the LINC complex and in maintaining its binding with SpdyA, thereby sustaining the integrity of the telomere-LINC complex at the NE. Therefore, the phosphorylation of SUN1 at S48 within the LINC complex at the non-PAR telomeres of the Y chromosome may be of particular significance in initiating Y-X NH desynapsis.

In conclusion, our study demonstrates that SpdyA plays a crucial role in mediating the NH desynapsis of the X-Y chromosomes at early pachynema. This SpdyA-governed process ensures X-Y desynapsis and proper X-Y loop-axis organization in the sex body. Equally importantly, our study points out that Y-X NH desynapsis and its subsequent pachytene X-Y loop-axis organization are distinct events, separate from MSCI, recombination, and sex body formation, thus representing a previously unknown aspect of pachynema progression. Further studies are required to investigate how X-Y chromosome compaction promotes pachynema progression, as defects in this process may contribute to sterility in some NOA patients.

## Methods

**Reagents and tools table**

| Reagent/resource | Reference or source | Identifier or catalog number |
| --- | --- | --- |
| **Experimental models** | | |
| HEK293T cells (*H. sapiens*) | ATCC | CRL-3216 |
| C57BL6/J (*M. musculus*) | CYAGEN | N/A |
| **Recombinant DNA** | | |
| pcDNA3.1(+) | ThermoFisher | V79020 |
| pcDNA3.1-Spdya^{WT}-HA | This study | N/A |
| pcDNA3.1-Spdya^{A125V}-HA | This study | N/A |
| pcDNA3.1-Cdk2-Myc | This study | N/A |
| pcDNA3.1-Sun1^{WT}-FLAG | This study | N/A |

| Reagent/resource | Reference or source | Identifier or catalog number |
|---|---|---|
| pcDNA3.1-Sun1$^{S48A}$-FLAG | This study | N/A |
| **Antibodies** | | |
| Mouse anti-SYCP3 | Abcam | ab97672 |
| Chicken anti-SYCP3 | This study | N/A |
| Guinea pig anti-SYCP1 | This study | N/A |
| Rabbit anti-SYCE1 | Proteintech | 11063-1-AP |
| Rabbit anti-TEX12 | Proteintech | 17068-1-AP |
| Rat anti-SPDYA | This study | N/A |
| Mouse anti-γH2AX | Millipore | 05-636 |
| Rabbit anti-γH2AX | Cell Signaling Technology | 9718 |
| Rabbit anti-SCML2 | Luo et al (2015) | N/A |
| Mouse anti-FK2 | Millipore | 04-263 |
| Human anti-ACA | Antibodiesinc | 15-234 |
| Rat anti-H1t | Lin et al (2024) | N/A |
| Mouse anti-MLH1 | BD Pharmingen™ | 51-1327GR |
| Mouse anti-TRF1 | Abcam | ab10579 |
| Rabbit anti-TERB1 | This study | N/A |
| Rabbit anti-MAJIN | This study | N/A |
| Rabbit anti-SUN1 | This study | N/A |
| Rabbit anti-KASH5 | This study | N/A |
| Rabbit anti-ANKRD31 | This study | N/A |
| Rabbit anti-p-SUN1(S48) | This study | N/A |
| Rabbit anti-HORMAD1 | Proteintech | 13917-1-AP |
| Rabbit anti-Lamin B1 | Proteintech | 12987-1-AP |
| Rabbit anti-Cleaved PARP (Asp214) | Cell Signaling Technology | 94885 |
| Rabbit anti-macroH2A1 | Sigma-Aldrich | ABE215 |
| Chicken anti-GFP | Abcam | Ab13970 |
| Mouse anti-DDX4 | Abcam | Ab27591 |
| Rabbit anti-SOX9 | Millipore | AB5535 |
| Mouse anti-ACTB | Proteintech | 66009-1-IG |
| Rabbit anti-CDK2 | Abcam | ab32147 |
| Rabbit anti-HA | Cell Signaling Technology | 3724 |
| Rabbit anti-DYKDDDDK (FLAG) | Cell Signaling Technology | 14793 |
| Mouse anti-MYC | Proteintech | 60003-2-IG |
| Rabbit anti-HIS | Proteintech | 10001-0-AP |
| **Oligonucleotides and other sequence-based reagents** | | |
| PCR primers | This study | Appendix Table S1 |
| **Chemicals, enzymes, and other reagents** | | |
| 2 × Rapid Taq Master Mix | Vazyme | P222-01 |
| Agarose | Biowest | BY-R0100 |
| N,N′-Octamethylenebis (2,2-dichloroacetamide) | Santa Cruz Biotechnology | sc-295819A |
| Retinoic acid | Sigma-Aldrich | R2625 |

| Reagent/resource | Reference or source | Identifier or catalog number |
|---|---|---|
| Corn oil | Sigma-Aldrich | C8267 |
| Tamoxifen | Sigma-Aldrich | T5648 |
| Paraformaldehyde | Sigma-Aldrich | 158127 |
| Bouin's Fixative Solution | Phygene | PH0976 |
| Prolong™ Diamond Antifade Mountant with DAPI | ThermoFisher | P36962 |
| VECTASHIELD antifade mounting medium with DAPI | Vector Laboratories | H-1200 |
| Y whole-chromosome probe | Empire Genomics | MCENY-10-RE |
| TelC-Cy3 probe | PNA BIO | F1002 |
| UltraPure™ 20x SSC Buffer | ThermoFisher | 15557044 |
| Transporter 5 | Polysciences | 26008 |
| MG132 | Selleckchem | S2619 |
| Cycloheximide | MedChemExpress | HY-12320 |
| Pierce™ IP lysis buffer | ThermoFisher | 87787 |
| Complete Protease Inhibitor | Roche | 4693159001 |
| Anti-HA Agarose | ThermoFisher | 26181 |
| Anti-DYKDDDDK (FLAG) Affinity Resin | ThermoFisher | A36801 |
| Protein A Agarose | Roche | 11719708001 |
| Protein G Agarose | Roche | 11719716001 |
| SurePAGE™, Bis-Tris, 10 × 8, 4–12%, procast gel | Genscript | M00652 |
| Skim milk | Oxoid | LP0031 |
| Ovalbumin | MedChemExpress | HY-W250978 |
| ATP | MedChemExpress | HY-B2176 |
| **Software** | | |
| ZEISS ZEN 3.8 | https://www.zeiss.com | |
| GraphPad Prism 10.0 | https://www.graphpad.com | |
| Image Lab Software | https://www.bio-rad.com | |
| ImageJ | https://imagej.net | |
| **Other** | | |
| Carl Zeiss LSM 900 inverted confocal microscope | Zeiss | LSM 900 |
| Bio-Rad ChemiDoc XRS+ Imager | Bio-Rad | |
| Illumina HiSeq PE150 | Illumina | |
| Illumina Novaseq X plus | Illumina | |
| TimsTOF Pro2 mass spectrometry | Bruker | |

## Mice

All mouse lines used in this study were generated in the C57BL/6J genetic background by Cyagen Biosciences (Suzhou, China) using the CRISPR/Cas9-mediated genome-editing system unless otherwise specified. Briefly, Cas9 mRNA, two gRNAs targeting specific

exons of the studied gene, and the donor vector containing the modified sequence were co-injected into fertilized mouse eggs to generate targeted knock-in offspring. The F0 founder mice were identified by PCR and DNA sequencing analysis and bred with wild-type mice to obtain F1 heterozygous mice. The F2 homozygous mice were obtained by in-crossing F1 heterozygous male and female mice. The primers used for genotyping are listed in Appendix Table S1. *Spdya^fl/fl* mice were generated by inserting two loxP sites flanking exon 2 of the *Spdya* gene. The *Ddx4-Cre^ERT2* mice were generated by inserting a "Cre^ERT2-P2A" cassette into exon 2 of the *Ddx4* gene (Appendix Fig. S13A). The specificity of the Ddx4-Cre^ERT2 fusion protein in mediating gene deletion in germ cells upon tamoxifen induction was validated by crossing the *Ddx4-Cre^ERT2* mouse line with *mT/mG* reporter mice (Appendix Fig. S13A–C). The *mT/mG* mice were kindly provided by Dr. Hua Zhang at China Agricultural University. As described previously (Muzumdar et al, 2007), upon Cre recombinase-mediated DNA deletion, the expression of the membrane-targeted red tdTomato fluorescence (mT) would be switched to that of the membrane-targeted green EGFP fluorescence (mG). After multiple rounds of crossing, *Spdya^fl/fl;Ddx4-Cre^ERT2* mice were obtained.

The *Trf1^fl/fl* mice were generated by inserting two loxP sites flanking exon 3 of the *Trf1* gene. The *Stra8-EGFPCre* mouse line, which expresses Cre recombinase driven by the endogenous *Stra8* promoter in germ cells, was obtained from Cyagen Biosciences (Stock No: C001283, Suzhou, China). The *Trf1^fl/fl* mice were first crossed to *Stra8-EGFPCre* mice and *Ddx4-Cre^ERT2* mice separately, and the resulting fertile *Trf1^fl/-* mice and *Trf1^fl/fl;Ddx4-Cre^ERT2* mice were then crossed with each other to generate *Trf1^fl/-;Ddx4-Cre^ERT2* mice.

The *Spdya^A125V* mice were generated by introducing a single nucleotide substitution (c.C374T) in the *Spdya* gene. The resulting p.A125V mutation in the mouse *Spdya* gene is equivalent to the p.A126V mutation in the human *SPDYA* gene. The *Spdya^HA/HA* mice were generated by inserting an HA tag downstream of the ATG start codon of the *Spdya* gene. The *Spdya^HA/HA* mice were fertile.

Mice were housed under controlled environmental conditions with free access to water and food, and illumination was on from 06:00 to 18:00. All experimental protocols were approved by the regional ethics committee of The University of Hong Kong-Shenzhen Hospital, Shenzhen, China.

### Human participants

The human study was approved by the Institutional Ethical Review Committee of Shanghai General Hospital, Shanghai Jiao Tong University School of Medicine (2020SQ199), Shanghai, China. Informed consents for the use of the clinical data, testicular tissue, and peripheral blood for research purposes were obtained from the donors.

For the initial cohort, we selected 1072 idiopathic NOA (iNOA) patients at the Department of Andrology, Urologic Medical Center, Shanghai General Hospital, Shanghai Jiao Tong University School of Medicine, Shanghai, China. Physical examination and repeated semen analysis after centrifugation were performed and analyzed according to the guidelines of the World Health Organization (6th edition). Known causal factors for male infertility, including

orchitis, cryprtochidism, varicocele, radiation, chemotherapy, and testicular cancer, were excluded. The genetic screening, including karyotype and Y chromosome microdeletions, was carried out in the NOA patients. The definition of the testis phenotype was based on multiple testis biopsies; one fragment was analyzed by the pathologist, who described the histological features, while the remaining fragments were used to search for spermatozoa by the embryologist. In the end, 318 subjects with maturation arrest were recruited in the present study to undergo WES (whole exome sequencing) and one was identified to harbor bi-allelic *SPDYA* mutation (P21492). One patient with obstructive azoospermia (OA) who had normal spermatogenesis but epididymis obstruction was enrolled as the positive control.

The proband (aged 30, male, Chinese) had been diagnosed with male infertility for 2 years, and routine semen analysis revealed complete azoospermia with normal volume. Physical examination revealed normal testicular volume in both sides (12 mL), non-dilated epididymis, and palpable vas deferens on both sides without anamnestic risk factors for obstruction. The proband had a normal 46, XY karyotype and no Y chromosome microdeletions. Reproductive hormone levels were within the normal range. The patient underwent a microsurgical testicular sperm extraction procedure, but no sperm were successfully retrieved. The clinical information of this patient is summarized in Appendix Table S2.

### WIN 18,446 and retinoic acid treatment

WIN 18,446 [i.e., N,N′-Octamethylenebis (2,2-dichloroaceta-mide)], which inhibits RA synthesis (Hogarth et al, 2013), was resuspended at 200 mg/ml in DMSO and further diluted with corn oil to a final concentration of 10 mg/ml. Fresh WIN 18,446 suspension was prepared directly before injection to avoid precipitation. RA was resuspended to 100 mM in DMSO and further diluted with corn oil to a final concentration of 4.16 mM. Following an established protocol (Romer et al, 2018), PD2 neonatal male mouse pups were injected *subcutaneously* (s.c.) with WIN 18,446 (100 mg/kg body weight) daily for 7 days to keep spermatogonia in an undifferentiated state (Fig. S2A). At PD9, mice were s.c. injected with a single dose of RA (12.5 mg/kg body weight) to induce synchronous spermatogonial differentiation and prophase I progression. The state of synchronous prophase I was validated by a series of immunostaining for SYCP3 and SYCP1 on testis sections collected at different days post RA restoration (Appendix Fig. S1B).

### Tamoxifen administration for specific targeting of pachytene spermatocytes in juvenile male mice

Tamoxifen was directly dissolved with corn oil to a final concentration of 5 mg/ml. To obtain *Spdya^cKO* pachytene spermatocytes, *Spdya^fl/fl;Ddx4-Cre^ERT2* male mice were injected i.p. with tamoxifen (20 mg/kg body weight) daily on 2 consecutive days at PD16 and PD17. Testes were collected at PD22, 5 days after the last tamoxifen injection. *Spdya^fl/fl* littermate mice received identical treatments and were used as controls.

To obtain early or mid-pachytene *Sync-Spdya^cKO* pachytene spermatocytes with a high frequency of *full Y–X NH synapsis*, synchronized *Spdya^fl/fl;Ddx4-Cre^ERT2* mice were i.p. injected with

tamoxifen (20 mg/kg body weight) for two consecutive days at RA7.25 and RA8.25 (i.e., PD16 and PD17). Testes were collected either at RA10.25 (i.e., PD19) or at RA11.25 (PD20) for analysis. To obtain late pachytene *Sync-Spdya^cKO* spermatocytes, synchronized *Spdya^fl/fl;Ddx4-Cre^ERT2* were i.p. injected with tamoxifen (20 mg/kg body weight) for 2 consecutive days at RA8.75 and RA9.75 (i.e., PD18 and PD19). Testes were collected at RA13 (i.e., PD22) for analysis. To obtain diplotene *Sync-Spdya^cKO* spermatocytes, synchronized *Spdya^fl/fl;Ddx4-Cre^ERT2* were i.p. injected with tamoxifen (20 mg/kg body weight) for 2 consecutive days at RA10 and RA11 (i.e., PD19 and PD20). Testes were collected at RA14.25 (i.e., PD23) for analysis. The synchronized *Spdya^fl/fl* littermate mice received identical treatments and were used as controls.

To obtain *Sync-Trf1^cKO* pachytene spermatocytes, synchronized *Trf1^fl/−;Ddx4-Cre^ERT2* and *Trf1^fl/fl* mice were i.p. injected with tamoxifen (20 mg/kg body weight) for 2 consecutive days at RA6 and RA7 (i.e., PD15 and PD16). Testes were collected at RA10 (i.e., PD19) for analysis.

## Antibody production

To examine the localization of SpdyA on chromosome spreads, polyclonal antibodies against mouse SpdyA (full length) were raised in rats by s.c. injection of 100–200 μg of purified TrxA-6×His-SpdyA on days 0, 15, 30, and 45. The rats were bled on day 54, and serum was prepared. The affinity column was prepared by coupling 1 mg purified TrxA-6×His-SpdyA protein to CNBr-activated Sepharose 4B (GE Healthcare). The antiserum was applied to the column, and the specific antibody was eluted with glycine HCl buffer (pH 2.5), followed by dialysis and concentration quantification. Protein purification, immunization of the animals and antibody affinity purification were performed by Dia-An Biotech, Inc. (Wuhan, China).

To characterize the phosphorylation state of the SUN1 protein, anti-pS48-SUN1 polyclonal antibodies were raised in rabbits immunized with the synthesized phosphorylated peptide LEPVFD (pS) PRMSR, a sequence corresponding to amino acids 42-53 of wild-type mouse SUN1, absorbed with non-phosphorylated peptide LEPVFDSPRMSR, and enriched with the phosphorylated peptide. Peptide synthesis, immunization of the animals and antibody affinity purification were done by YouKe Biological Technology Co. (Shanghai, China).

## Indirect immunofluorescence (IF) analyses

Mouse spermatocyte chromosome spreads were prepared as described previously (Peters et al, 1997). Briefly, after removing the tunica albuginea of the testis, the seminiferous tubules were carefully separated with a pair of fine forceps, subjected to a hypotonic buffer (30 mM Tris-HCl, 17 mM trisodium citrate dihydrate, 5 mM EDTA, 50 mM sucrose, 1 mM PMSF, 2.5 mM DTT, pH 8.2–8.4) for 25 min, and immediately transferred to 100 mM sucrose buffer for 5 min. The cell suspension obtained by gently squeezing the tubules with forceps was added to slides containing fixative buffer [1% paraformaldehyde (PFA), 0.15% Triton X-100, pH 9.2]. The air-dried spreads were then blocked with 5% BSA and incubated with primary antibodies at 4 °C overnight.

For IF analyses of testis sections, testes were fixed in 4% PFA for 24 h, dehydrated, and embedded in paraffin. After rehydration, the 5-μm sections were subjected to antigen retrieval by heating in Tris-EDTA buffer (pH 9.0) for 20 min. The cooled slides were then subjected to 5% BSA blocking and primary antibody staining. Primary antibodies were detected with Alexa Fluor 488-, 594- or 647-conjugated secondary antibodies. The slides were washed with PBS and mounted using Prolong™ Diamond Antifade Mountant with DAPI (Invitrogen). The primary antibodies used for IF analyses are listed in Appendix Fig. S3.

## Immunofluorescence in situ hybridization (immuno-FISH)

The immuno-FISH assay to detect Y chromatin loops was carried out using a Y whole-chromosome probe (Empire Genomics). After IF, spread slides were fixed in 1% PFA (pH 9.2) for 10 min, washed with PBS, and dehydrated with 70%, 85% and 100% ethanol. The air-dried slides were heated in the prewarmed denaturation buffer [70% formamide, 2× saline sodium citrate (SSC), pH 7.0] at 73 °C for 5 min then washed, and dehydrated again with 70%, 85%, and 100% ethanol. The pre-denatured Y chromosome-specific probe was applied to the air-dried and pre-denatured slides and allowed to hybridize in a humid chamber at 37 °C for a minimum of 16 h. Four washes were conducted at 42 °C in 4× SSC with 0.2% Tween 20, each lasting 5 min.

Immuno-FISH to detect telomeres in testis sections was carried out based on a FISH protocol with minor modifications (Scherthan, 2017). In brief, after incubation with secondary antibodies, testis sections were washed, dehydrated, with 70%, 85%, and 100% ethanol and air-dried. After denaturation at 85 °C for 10 min, the sections were hybridized for 2 h at 37 °C with a Cy3-labeled (CCCTAA)$_3$ PNA probe (PNA BIO) in hybridization buffer [20 mM Tris, pH 7.4, 60% formamide, 0.5% blocking reagent (Roche)]. The sections were washed sequentially in 2× SSC with 0.1% Tween 20 at 55 °C (twice) and with 2× SSC alone at room temperature (twice) for 5 min each time.

Immunolabeled chromosome spreads and tissue section samples were imaged with a Carl Zeiss LSM 900 inverted confocal microscope driven by Zeiss Efficient Navigation (ZEN) 3.2 software for Windows 10 (64 bits). Airyscan processing and Orthological projection functions were applied to prepare a stacked image for analysis using ZEN 3.2.

## Immunoblotting and co-immunoprecipitation (Co-IP)

For transfection into HEK293T cells, the cDNAs encoding SpdyA^WT-HA, SpdyA^A125V-HA, CDK2-Myc, SUN1^WT-FLAG, and SUN1^S48A-FLAG were constructed into the *pcDNA3.1* vector (referred as to *pcDNA3.1-Spdya^WT-HA*, *pcDNA3.1-Spdya^A125V-HA*, *pcDNA3.1-Cdk2-Myc*, *pcDNA3.1-Sun1^WT-FLAG*, and *pcDNA3.1-Sun1^S48A-FLAG*) and transfected using Transporter 5 (Polysciences). After 48 h, cells were washed three times in cold PBS and lysed in IP lysis buffer (Pierce) supplemented with Complete Protease Inhibitor (Roche) on ice for 20 min. The extracts were then centrifuged at 13,000×*g* for 20 min at 4 °C. The supernatants were used for immunoblots. For Co-IP experiments, cell lysates were incubated with anti-HA agarose (Thermo Scientific Pierce) or anti-

FLAG (DYKDDDDK) Affinity Resin (Thermo Scientific Pierce) or with anti-Myc antibody followed by Protein A/G Agarose (Roche) at 4 °C on a tube rotator overnight. Precipitates were then washed with cold Tris-buffered saline with 0.05% Tween 20, resuspended in SDS-loading buffer, and boiled at 95 °C for 5 min. Equal amounts of proteins were loaded and electrophoresed on 4–12% SurePAGE precast protein gels (Genscript) and transferred onto polyvinylidene fluoride membranes (Millipore). The membranes were blocked with 5% skim milk (Oxoid) at room temperature for 1 h and probed with specific antibodies. Immunoreactive bands were detected and analyzed with a Bio-Rad ChemiDoc XRS+ Imager. β-actin was used as the internal loading reference. Each immunoblot was repeated at least three times using independent biological replicates.

## MG132 treatment and cycloheximide chase assays

SpdyA protein overexpressed in HEK293T cells is degraded by the proteasome (Al Sorkhy et al, 2009). To assess the ubiquitination-mediated degradation of the SpdyA protein 4 h after transfection of HEK293T cells with *pcDNA3.1-Cdk2-Myc* and *pcDNA3.1-Spdya^WT^-HA* or *pcDNA3.1-Spdya^A125V^-HA* plasmids, the cultures were treated with 5 μM MG132 for 12 h and cell lysates were analyzed by immunoblots. To assess the rate of protein degradation, 48 h after transfection of HEK293T cells with *pcDNA3.1-Cdk2-Myc*, and *pcDNA3.1-Spdya^WT^-HA*, or *pcDNA3.1-Spdya^A125V^-HA* expression vectors, the cultures were treated with cycloheximide (50 μg/ml) for 0–24 h. Immunoblots were used to detect the Spdya^WT^-HA or Spdya^A125V^-HA protein levels.

## Bulk RNA sequencing and data analyses

The bulk RNA sequencing was performed by Novogene Corporation Inc. Total RNA was purified from RA11.50 *Sync-Spdya^fl/fl^* and *Sync-Spdya^cKO^* testes (mid-late pachytene testes) using Trizol reagent (Invitrogen) according to the supplier's instructions. The RNA quality and quantity were evaluated via Bioanalyzer (Agilent) and Qubit (Life Technologies), respectively. The assay was performed with an Illumina HiSeq PE150 (Novogene). Normalization was performed using Novomagic software (Novogene) to generate Fragments Per Kilobase Million (FPKM). Protein-coding genes having an average FPKM > 1 in both groups were included in the transcriptional profile comparison. Differential expression analysis was performed using DESeq2, and genes with FDR-adjusted *P* value < 0.05 and |log$_2$ Fold change| >1 were considered to be differentially expressed.

## Single-cell library preparation and sequencing

The single-cell library preparation was carried out by Novogene Corporation Inc. Briefly, RA11.75 *Sync-Spdya^fl/fl^* and *Sync-Spdya^cKO^* testes (mid-late pachytene testes) from 5 mice per genotype were pooled and subjected to enzyme digestion, and the sorted single-cell suspension was loaded into Chromium microfluidic chips with 3′ (v3) chemistry and barcoded with a 10× Chromium Controller (10× Genomics). RNA from the barcoded cells was subsequently reverse-transcribed using a Chromium Single Cell 3′ (v3) reagent kit (10× Genomics). The library was sequenced using the Illumina Novaseq X plus system.

## Single-cell RNA sequencing data analysis

The gene expression matrices were obtained using Cell Ranger (version 4) and were filtered using Seurat v4.0.4. Doublets were removed using DoubletFinder. Bioinformatics analyses, such as UMAP clustering and differential gene expression analysis, were conducted using Omicsmart, an interactive online platform designed for real-time data analysis (http://www.omicsmart.com).

## TEM analysis

Mice were sacrificed, and the testes were collected and transferred into an Eppendorf tube with fresh TEM fixative (Servicebio) for 2 h at RT and then stored at 4 °C for transportation. Resin embedding, sectioning, and TEM imaging were performed by Wuhan Servicebio Technology Co., Ltd. (Wuhan, China). As previously described (Lin et al, 2024), the tissues were post-fixed with 1% OsO$_4$ (Ted Pella Inc), dehydrated, and embedded in EMBed 812. After polymerization, the resin blocks were cut into 60–80 nm ultrathin sections using an ultramicrotome (Leica UC7). The sections were then placed on 150 mesh copper grids, stained with 2% uranium acetate and 2.6% lead citrate, and imaged by TEM (HITACHI, HT7800).

## Phosphoproteomic analysis

The data-independent acquisition (DIA)-based phosphoproteomic analysis was performed by Shanghai Luming Biological Technology Co., LTD (Shanghai, China). RA10.25 *Sync-Spdya^fl/fl^* and *Sync-Spdya^cKO^* testes (2 days after the second tamoxifen injection) were ground in liquid nitrogen and homogenized on ice in lysis buffer (Beyotime) with Phosphatase inhibitor cocktail (Beyotime). After centrifugation, the supernatant was collected, and the protein concentration was quantified with a BCA kit. After trypsin digestion, the peptides were desalted on a Sep-Pak C18 Cartridge (Waters), vacuum-dried, and reconstituted in 50% acetonitrile with 0.1% formic acid (Thermo Scientific) according to the manufacturer's instructions. The phosphopeptides were enriched using a High-Select Fe-NTA Phosphopeptide Enrichment Kit (Thermo Scientific). Each sample was mixed with the indexed retention time (iRT) internal-standard mixture (Biognosys) at a ratio of 20:1. Peptide samples were analyzed on a nanoElute liquid chromatography system (Bruker) coupled with a TimsTOF Pro2 mass spectrometry (Bruker). The resulting DIA raw data were analyzed by searching the constructed spectral library with a Spectronaut Pulsar 18.7 (Biognosys). The main parameters of the software–Precursor Qvalue cutoff and Protein Qvalue cutoff–were both set to 0.01.

## SUN1 protein kinase assay

The production and purification of TrxA-6×His-SUN1$_{1-217}$ (WT) or TrxA-6×His-SUN1$_{1-217}$ (S48A) recombinant proteins were carried out by Mabnus Biotech, Inc. (Wuhan, China). Briefly, the cDNAs encoding mouse SUN1 amino acids 1–217, either WT or the S48A mutant, were subcloned into the *pET-32a* (+) vector and expressed in *Escherichia coli*. The cells were cultured at 37 °C and then induced with 1 mM IPTG for 12 h at 18 °C. Following cell

collection, sonication, and centrifugation, the soluble fractions were purified using Ni-NTA Agarose. The purified proteins were subsequently dialyzed in EB buffer containing 80 mM β-glycerophosphate, 20 mM EGTA, and 15 mM $MgCl_2$, pH 7.3. The protein purity was analyzed with 4–20% SDS-PAGE and Coomassie brilliant blue staining.

To determine the endogenous kinase activity of the SpdyA complex in pachytene spermatocytes, fresh testes from PD17 $Spdya^{HA/HA}$ and $Spdya^{+/+}$ mice were decapsulated and lysed in $EBN^{++}$ buffer containing 80 mM β-glycerophosphate (pH 7.3), 20 mM EGTA, 15 mM $MgCl_2$, 150 mM NaCl, 1 mg/ml ovalbumin, and 0.5% NP-40 supplemented with Complete Protease Inhibitor on ice for 30 min. The extracts were then centrifuged at 13,000×$g$ for 30 min at 4 °C. The supernatants were used for immunoprecipitation.

A total of 3 mg of the testis protein lysates from each group were incubated with Anti-HA Agarose at 4 °C on a tube rotator overnight. The immunoprecipitation beads were then washed twice in cold $EBN^{++}$ buffer and twice in $EB^{++}$ buffer ($EBN^{++}$ buffer without NaCl and NP-40) and then incubated with 3 μg TrxA-6×His-SUN1$_{1-217}$ (WT) or TrxA-6×His-SUN1$_{1-217}$ (S48A) peptides, 15 mM ATP (MedChemExpress), and 10 mM DTT in $EB^{++}$ buffer for 1 h at 37 °C. Reactions were terminated in 1× SDS-loading buffer, boiled for 5 min, and subjected to immunoblotting analyses.

## Statistical analysis

Statistical analyses were conducted using GraphPad Prism software version 10.0 (GraphPad Software Inc.). For statistical analysis of the differences between two groups, two-tailed unpaired Student's $t$ tests were used. For statistical analyses of data presented as scatter plots and c-PARP$^+$ cell ratio comparisons, nonparametric two-tailed Mann–Whitney tests were used. Investigators were not blinded to mouse genotypes or cell genotypes during the experiments.

## Data availability

The bulk RNA-Seq data have been deposited to the Gene Expression Omnibus (GEO) repository under accession number GSE276678. The scRNA-seq raw data are available from the Genome Sequence Archive in the National Genomics Data Center (https://www.cncb.ac.cn/) with accession number CRA026315 from project PRJCA040919. Mass spectrometry proteomics and phosphoproteomic data have been deposited with the ProteomeXchange Consortium (https://proteomecentral.proteomexchange.org) via the iProX partner repository (Chen et al, 2022; Ma et al, 2019) with the dataset identifier PXD056108.

The source data of this paper are collected in the following database record: biostudies:S-SCDT-10_1038-S44318-025-00528-8.

## Peer review information

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

## Acknowledgements

This work was supported by grants from the Shenzhen-Hong Kong-Macau Type C Project (SGDX20230821091501011 to KL), the Shenzhen Science and Technology Program (KQTD20190929172749226 to KL and WSBY), the National Natural Science Foundation of China and the Swedish Research Council Collaboration Research Program (NSFC-VR 8211101255 to KL), the National Key Research and Development Program of China (2022YFC2702701 to ZL) and the Sanming Project of Medicine in Shenzhen (SZSM 202211014 to WSBY). We thank many colleagues and researchers for their constructive discussions.

## Author contributions

**Dongteng Liu**: Conceptualization; Data curation; Formal analysis; Validation; Investigation; Visualization; Methodology; Writing—original draft; Project administration; Writing—review and editing. **Yuxiang Zhang**: Resources; Investigation; Writing—review and editing. **Dongliang Li**: Formal analysis; Investigation; Visualization. **Binjie Jiang**: Formal analysis; Validation; Investigation. **Xudong Zhao**: Investigation; Methodology. **Yanyan Li**: Validation; Investigation. **Zexiong Lin**: Investigation. **Yu Zhao**: Investigation. **Zhe Hu**: Investigation; Methodology. **Shuzi Deng**: Investigation. **Zheng Li**: Funding acquisition. **Haonan Lu**: Resources. **Karen KL Chan**: Resources; Project administration. **William SB Yeung**: Funding acquisition; Project administration. **Philipp Kaldis**: Methodology. **Chencheng Yao**: Resources; Investigation. **Hengbin Wang**: Writing—review and editing. **Louise T Chow**: Conceptualization; Project administration; Writing—review and editing. **Kui Liu**: Conceptualization; Resources; Data curation; Supervision; Funding acquisition; Investigation; Visualization; Methodology; Writing—original draft; Project administration; Writing—review and editing.

Source data underlying figure panels in this paper may have individual authorship assigned. Where available, figure panel/source data authorship is listed in the following database record: biostudies:S-SCDT-10_1038-S44318-025-00528-8.

## Disclosure and competing interests statement

The authors declare no competing interests.

# Expanded View Figures

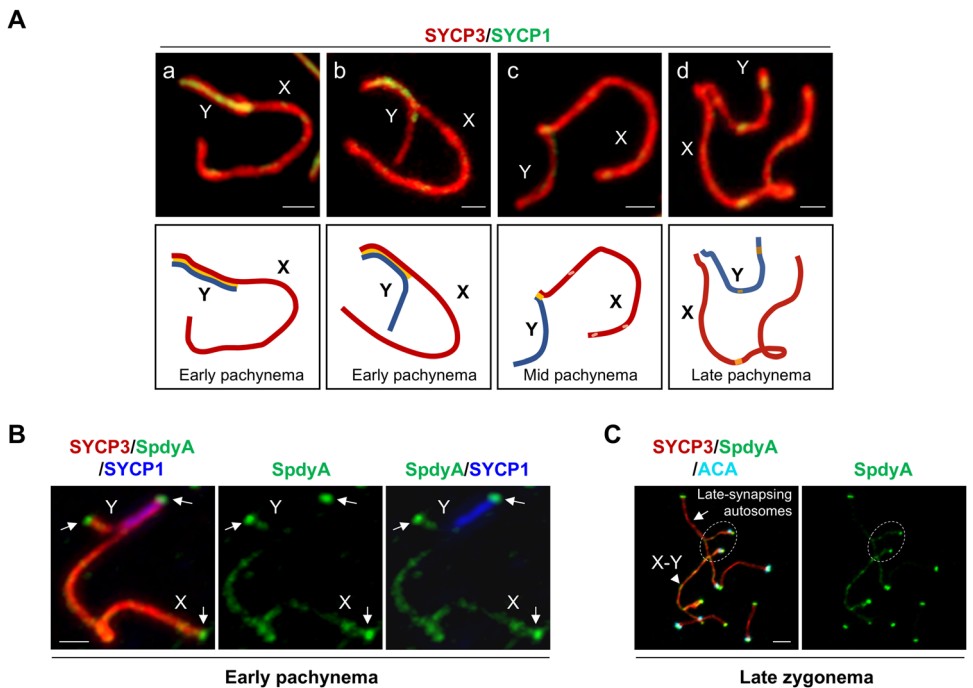

**Figure EV1.   Dynamic configuration changes of X and Y chromosomes and SpdyA localization at sex chromosome axes and telomeres during pachynema.**

(**A**) Illustration of the dynamic changes in the X and Y chromosome configuration during pachynema. X and Y chromosomes were immunostained for SYCP3 (red) and SYCP1 (green). (a) At early pachynema, the X and Y chromosomes exhibit extensive side-by-side pairing, with the Y chromosomal axis fully aligned along the X chromosomal axis and with synaptonemal complex (SC) formation occurring between the aligned regions. This configuration is transient and rarely observed. (b) As pachynema progresses, the SC between the X and Y chromosomes gradually shortens. (c) By mid-pachynema, the SC becomes restricted to the PAR. (d) At late pachynema, desynapsis occurs and a chiasma forms, resulting in an "end-to-end" attachment between the X and Y chromosomes. Scale bars, 1 μm. (**B**) The localization of SpdyA at telomeres (arrows) and at the unsynapsed axes of X and Y chromosomes at early pachynema. X and Y chromosomes were immunostained for SYCP3 (red), SYCP1 (blue) and SpdyA (green). Scale bar, 1 μm. (**C**) SpdyA localization at the unsynapsed region (dashed circle) of late-synapsing autosomes at late zygonema. Scale bar, 2 μm.

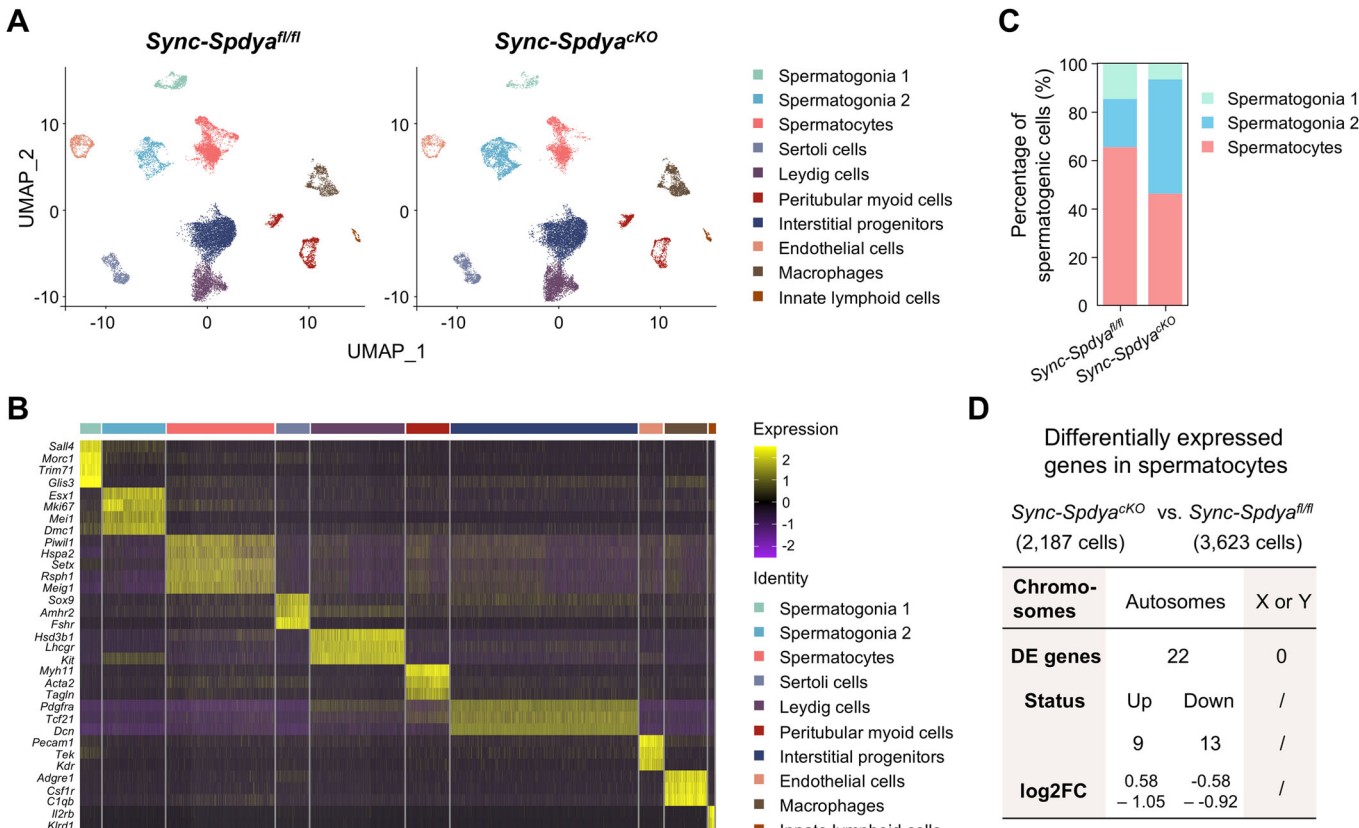

**Figure EV2.   Single-cell RNA-seq analysis revealed no differential expression of sex chromosome-linked genes in mid-late *Sync-Spdya^cKO* pachytene spermatocytes.**

(A) UMAP and clustering analysis of single-cell transcriptome data from testicular cells collected at RA11.75 from tamoxifen-treated *Spdya^fl/fl* and *Spdya^fl/fl;Ddx4-Cre^ERT2*
mice. Each dot represents a single cell, colored according to its cluster identity. UMAP, Uniform Manifold Approximation and Projection. (B) Heatmap showing expression
of marker genes for 10 identified cell types (right). The pachytene cell marker *Hspa2* is highly expressed in spermatocytes. Spermatogonia 1 and 2 likely represent
undifferentiated and differentiating spermatogonia, respectively, based on their marker gene profiles. (C) Proportion of the three types of spermatogenic cell types.
(D) Differential expression analysis of 2187 *Sync-Spdya^cKO* and 3623 *Sync-Spdya^fl/fl* pachytene spermatocytes. DE genes, differentially expressed genes; Log2FC, log$_2$ fold
change. Cutoff criteria: |log2FC| ≥ 0.58 and adjusted *P* value < 0.05.

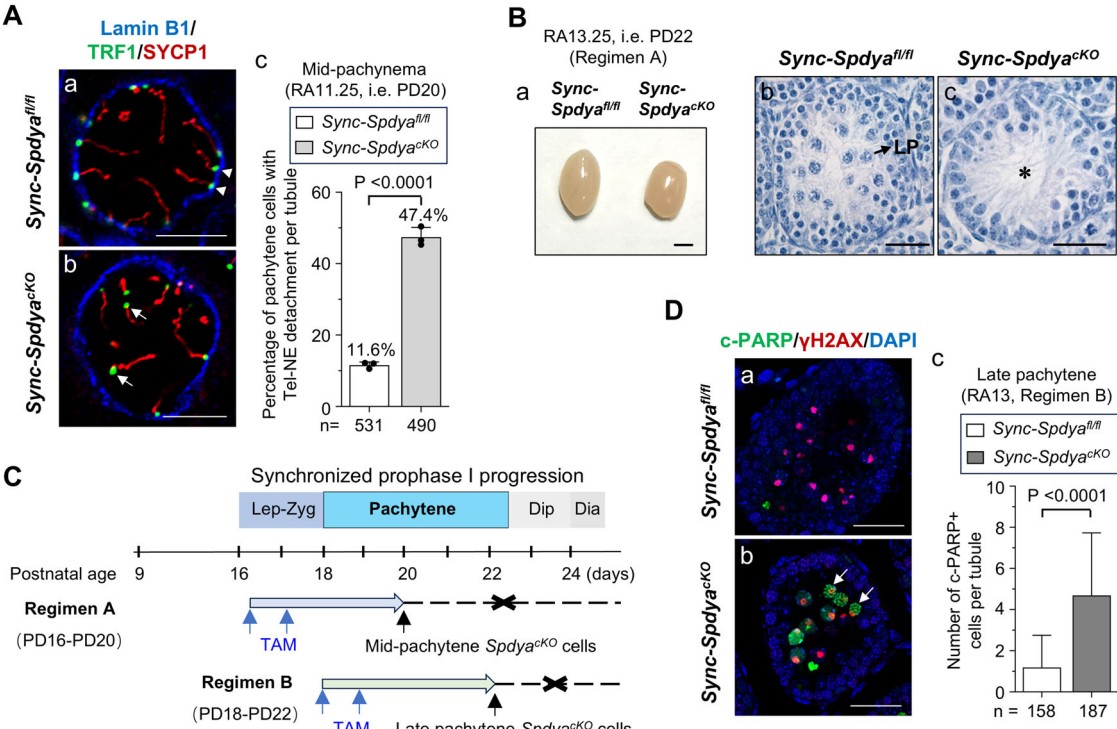

**Figure EV3.  SpdyA is essential for spermatocyte survival throughout pachynema.**

(A) Compared to *Sync-Spdya^fl/fl* cells (a), mid-pachytene (RA11.25) *Sync-Spdya^cKO* cell (b) exhibited severe Tel-NE detachment (arrows). Testis sections were immunostained for TRF1 (green), SYCP1 (red) and Lamin B1 (blue). Arrowheads indicate Tel-NE attachment. Scale bars, 5 µm. (c) Percentages of pachytene cells with Tel-NE detachment in mid-pachytene *Sync-Spdya^fl/fl* and *Sync-Spdya^cKO* testes at RA11.25. "*n*" represents the total number of pachytene spermatocytes scored from three mice per genotype. (B) Following tamoxifen regimen shown in Fig. 2A, the *Sync-Spdya^cKO* testes collected at RA13.25 were smaller in size and lacked late pachytene spermatocytes. LP, late pachytene spermatocyte; asterisk indicates loss of spermatocytes. Scale bars, 30 µm. (C) Summary of tamoxifen treatment regimens used to obtain mid- and late pachytene *Sync-Spdya^cKO* spermatocytes. Regimens A and B correspond to those shown in Figs. 2A and 3A, respectively. Dashed lines with crosses indicate stages at which *Sync-Spdya^cKO* spermatocytes could not be obtained. (D) Following Regimen B, late pachytene *Sync-Spdya^cKO* spermatocytes collected at RA13 exhibited apoptosis. Testis sections were immunostained for c-PARP (green) and γH2AX (red). Nuclei were counterstained with DAPI. Scale bars, 30 µm. "*n*" represents the total number of seminiferous tubules scored from three mice per genotype. Data information: All values are presented as the mean ± SD. Statistical analyses were performed using two-tailed Student's *t* test in (A) with *P* = 0.000025, and Mann–Whitney test in (D) with *P* < 0.000000000000001.

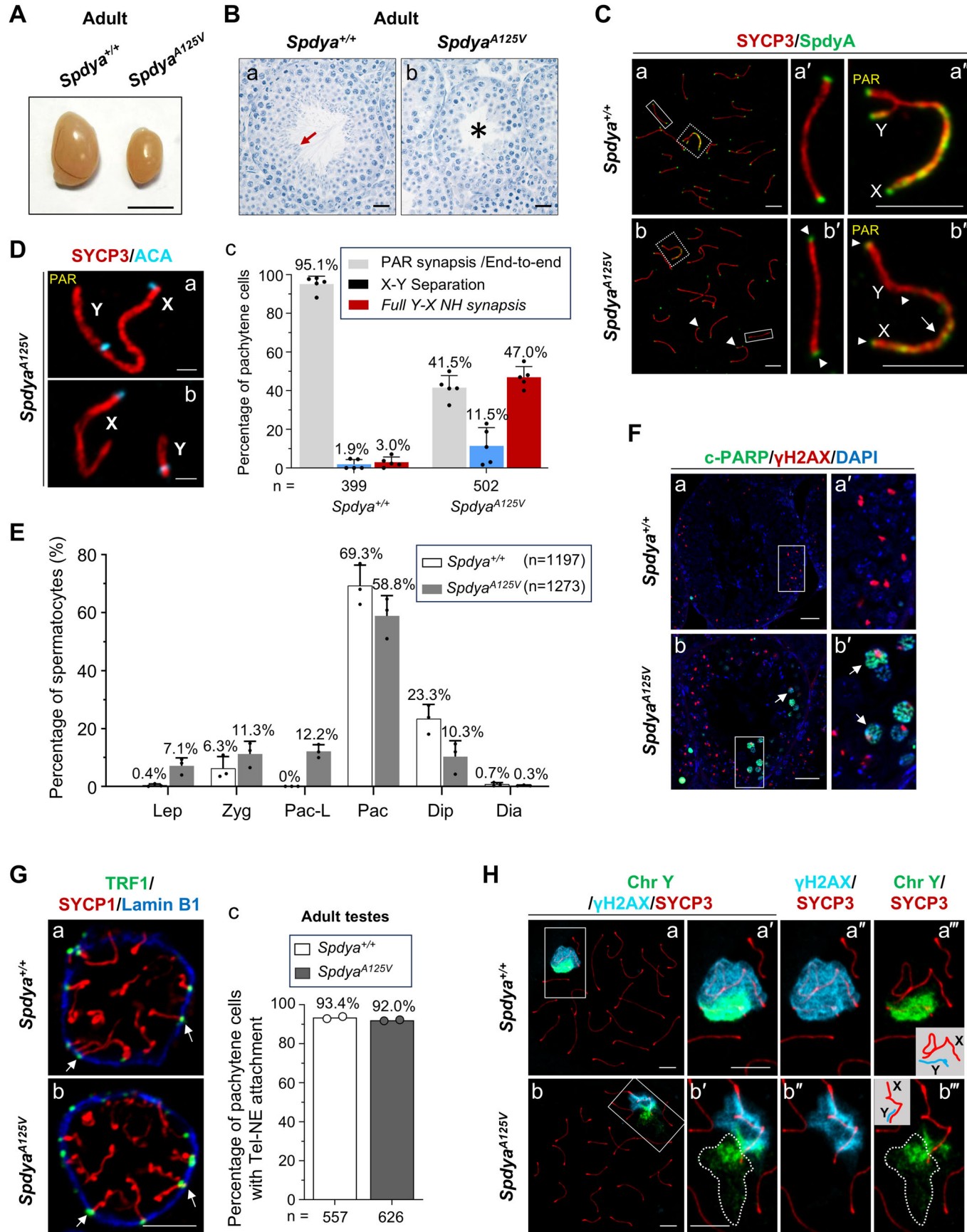

◀

**Figure EV4. Adult *Spdya^A125V* mice is a hypomorphic-SpdyA model in which pachytene spermatocytes exhibited *full Y-X NH synapsis* and apoptosis.**

(A) The size of testis was smaller in the 3-month-old (adult) *Spdya^A125V* mice. Scale bar, 5 mm. (B) Histological analyses of testes from adult *Spdya^+/+* (a) and *Spdya^A125V* (b) mice. Red arrow indicates great amounts of elongated spermatids in *Spdya^+/+* seminiferous tubule, but they were difficult to find in *Spdya^A125V* seminiferous tubule (asterisk). Scale bars, 30 μm. (C) SpdyA signal intensity was strikingly reduced at telomeres (b–b″, arrowheads) and sex chromosome axes (b″, arrow) in *Spdya^A125V* pachytene spermatocyte. Magnified views (a′–a″, b′–b″) are indicated by solid and dashed rectangles in (a–b). Scale bars, 5 μm. (D) Numbers of *Spdya^A125V* pachytene cells exhibited *full Y-X NH synapsis* (a) and X-Y separation (b). Scale bars, 1 μm. (c) Percentages of pachytene cells with different X-Y configurations in testes from *Spdya^+/+* and *Spdya^A125V* adult male mice. (E) Analyses of meiotic stages in spermatocytes from adult *Spdya^+/+* and *Spdya^A125V* mice, based on immunostaining of chromosome spreads. Lep, leptotene; Zyg, zygotene; Pac-L, pachytene-like; Pac, pachytene; Dip, diplotene; Dia, diakinesis spermatocytes. (F) IF analyses of *Spdya^+/+* (a–a′) and *Spdya^A125V* (b–b′) testis sections immunostained for c-PARP (green) and γH2AX (red). Solid rectangles indicate areas magnified in (a′–b′). Arrows indicate apoptotic spermatocytes. Scale bars, 30 μm. (G) IF analyses of *Spdya^+/+* and *Spdya^A125V* testis sections with Lamin B1 (blue), TRF1 (green) and SYCP1 (red). Arrows indicate Tel-NE attachment. Scale bars, 5 μm. (c) Percentages of *Spdya^+/+* and *Spdya^A125V* pachytene cells with intact Tel-NE attachment. (H) Sex chromosomes in late pachytene *Spdya^A125V* spermatocytes with *full Y-X NH synapsis* displayed disrupted loop-axis organization. Chromosome spreads were subjected to immuno-FISH staining for γH2AX (light blue), Chr Y (green) and SYCP3 (red). Magnified views (a′–a‴, b′–b‴) are indicated by solid rectangles in (a, b). White dashed line-enclosed area indicates Y chromatin signals located outside the γH2AX area. Scale bars, 5 μm. Data information: In (D, E), values are presented as the mean ± SD; In (G), data are presented as the mean with individual values. "n" represents the total number of pachytene spermatocytes scored per genotype: from five mice in (D), from three mice in (E), and from two mice in (G).

