## [Peer Review File · The EMBO Journal]

Speedy A governs non-homologous XY chromosome desynapsis as a unique prerequisite for XY loop-axis organization

Dongteng Liu, Yuxiang Zhang, Dongliang Li, Binjie Jiang, Xudong Zhao, Yanyan Li, Zexiong Lin, Yu Zhao, Zhe Hu, Shuzi Deng, Zheng Li, Haonan Lu, Karen Chan, William Shu Biu Yeung, Philipp Kaldis, Chencheng Yao, Hengbin Wang, Louise T. Chow, and Kui Liu

Corresponding author(s): Kui Liu (kliugc@hku.hk), Louise T. Chow (ltchow@uab.edu), Dongteng Liu (liudt@hku-szh.org)

Review Timeline:

Submission Date:	8th Feb 25
Editorial Decision:	5th Mar 25
Revision Received:	18th Jun 25
Accepted:	23rd Jul 25

Editor: Hartmut Vodermaier

Transaction Report:

Prof. Kui Liu
The University of Hong Kong
Department of Obstetrics and Gynecology, Li Ka Shing Faculty of Medicine
Hong Kong Special Administrative Region of the People's Republic of China

5th Mar 2025

Re: EMBOJ-2025-120432-T
Speedy A governs non-homologous XY desynapsis as a unique prerequisite for XY loop-axis organization

Dear Dr. Liu,

Thank you for submitting your manuscript on Speedy A in non-homologous XY desynapsis to The EMBO Journal. We have now received reports from two expert referees, copied below for your information. As you will see, both referees consider this work interesting and potentially important, but they also raise a number of major concerns with the analyses at this stage. These include unclear experimental details, issues with the RNA-seq data, inappropriate control/comparison conditions, as well as unexpected/discrepant outcomes lacking satisfactory explanation. Furthermore, there are a number of points regarding the presentation of background and findings both in the text and in the figures.

In this light, I would like to give you a chance to revise the manuscript in response to the reviews - but I must emphasize that the work will only become suitable for EMBO Journal publication if these criticisms could be satisfactorily clarified during a single major revision round. I would therefore encourage you to contact me with a revision plan and preliminary point-by-point response already during the early stages of your revision work, so that we could discuss if and how the main points could best be resolved. I would also be happy to extend the revision time beyond the regular three months if this should be needed. Our 'scooping protection' (meaning that competing work appearing elsewhere in the meantime will not affect our considerations of your study) would of course remain valid also throughout such an extension.

Detailed information on preparing, formatting and uploading a revised manuscript can be found below and in our Guide to Authors. Thank you again for the opportunity to consider this work for The EMBO Journal, and I look forward to hearing from you in due time.

Yours sincerely,

Hartmut Vodermaier

9) To facilitate reproducibility and cross-laboratory adoption of methodologies, please structure the Materials & Methods section as outlined in our guide to authors, including a completed Reagents and Tools Table that can be downloaded from our author guidelines as well (<https://www.embopress.org/page/journal/14602075/authorguide#structuredmethods>).

10) Digital image enhancement is acceptable practice, as long as it accurately represents the original data and conforms to community standards. If a figure has been subjected to significant electronic manipulation, this must be clearly noted in the figure legend and/or the 'Materials and Methods' section. The editors reserve the right to request original versions of figures and the original images that were used to assemble the figure. Finally, we generally encourage uploading of numerical as well as gel/blot image source data; for details see: embopress.org/page/journal/14602075/authorguide#sourcedata

At EMBO Press, we ask authors to provide source data for the main manuscript figures. Our source data coordinator will contact you to discuss which figure panels we would need source data for and will also provide you with helpful tips on how to upload and organize the files.

Revision to The EMBO Journal should be submitted online within 90 days, unless an extension has been requested and approved by the editor; please click on the link below to submit the revision online before 3rd Jun 2025:
Link Not Available

If you choose to alternatively have this study further considered by another EMBO Press publication, please use the following hyperlink to directly transfer the manuscript, optionally with inclusion of referee reports and identities:
Link Not Available

Referee #1:

General summary:

This paper by Liu et al extends our understanding of the role of Speedy A in male mouse meiosis. Apart from its previously characterized function in mediating telomere attachment to the nuclear envelope during leptotema, the authors have identified a later-stage requirement for Speedy A for the desynapsis of the non-homologous regions of chromosomes X and Y during pachynema of male meiosis. They carefully dissect this role, showing that the failed desynapsis of the X-Y non-homologous region does not affect MSCI or crossover formation, but instead disrupts the chromatin loop shortening that normally occurs on the Y chromosome at late pachynema. They continue to show that this role of Speedy A is mediated by its phosphorylating (via CDK2) SUN1 protein at S48, revealing a new role for the protein. This also adds new insight for the role of Y-X NH desynapsis during male mouse meiosis. Overall, the study is carried out carefully, with attention to detail and many important findings.

Plenty of overlapping evidence to strengthen points is given, and the manuscript's message is clear. However, there are some concerns that need to be addressed before publication.

Major concerns:

1. What is the fertility status of male mice with conditional deletion of Speedy A? Please comment on this in the text. Could you collect sperm from a later timepoint to perform IVF experiments?
2. To evaluate the state of MSCI (establishment of which is based on the asynapsed nature of the X and Y chromosomes) the authors performed RNAseq at their PD20 model of synchronized spermatogenesis. However, there are several concerns regarding this analysis:
 - a. first, it's unclear whether correlation coefficients are performed on all genes or the most variable ones
 - b. the number of total genes is very low for a mouse bulk RNAseq experiment (11,636), compared to an expected number of ~20000.
 - c. what were the 34 DE genes identified in the analysis? Is there anything interesting about them?
 - d. the comparison is made on bulk testis tissue, so includes all somatic cells of the testis, which express normal levels of X/Y transcripts. This makes interpretation about sex-gene expression in germ cells difficult to extract. The tamoxifen/Cre-mediated mutation, despite being quite efficient, can be quite variable (reported 68.1% in not-synchronized spermatocytes, but 92.3% in synchronized PD20 spermatocytes). Given that this might be different in each tamoxifen injection event, one cannot be certain as to the percentage of cells that are actually mutant in this experiment. To make the claim that MSCI is normal, single-cell RNAseq could be performed to ensure that only mutated cells are taken into account (judged by the lack of Speedy A transcript). Alternatively, RNA FISH for X and Y transcripts can be performed on chromosome spreads in conjunction with Speedy A IF, to only measure mutated cells. If the authors cannot perform this experiment, they should try reducing their expression fold-change cut-offs to a lower value (e.g. $\log_2FC=0.5$), and reassess expression.
3. Cell death is measured by c-PARP staining on testis sections. This is only commented as "many c-PARP positive pachytene cells" (line 268), but no quantification is provided. Does deletion of Speedy A obstruct progression to diplotene and further? Are any spermatids observed at later timepoints? Are these (if any) spermatids aneuploid?
4. When harvesting testes at PD20 to look at mid-pachynema, the authors inject tamoxifen at PD16/17, which in their synchronized model of spermatogenesis corresponds to leptotene/zygotene. However, to look at late pachynema (harvesting at PD22) they inject tamoxifen at days PD18/19, which corresponds to late zygotene/early pachynema. Therefore, mutation of the gene happens at different points during meiotic progression and can affect the results. All experiments performed at PD22 should be consistent with the PD20 experiments and should involve deletion at a similar timeframe. This is relevant because conclusions are drawn regarding the continuity of the process based on data from both timepoints. For example, to provide evidence that late-pachynema spermatocytes do not exhibit desynapsis (and therefore the persistence of the Y-X NH synapsis lasts from early to late pachynema), they carry out quantification of synapsed X-Y NH regions at PD22. However, this does not prove that the spermatocytes observed at PD20 would behave the same if left for 2 more days without the protein, and more importantly it does not prove that PD20 spermatocytes are not delayed and would not resolve the Y-X NH synapsis by PD22. Similarly, to look at early pachynema (PD19 collection), tamoxifen injections are carried out a day earlier than those for PD20 collections. It's unclear to us why these different timings were used; the authors should explain.
5. Given that in this mutant the X and Y undergo persistent synapsis, and that part of the Y doesn't accumulate MSCI-associated repressive marks, it is puzzling that the Y doesn't escape MSCI. The authors should address this in their discussion.

Minor comments:

Figure legends are long and contain extensive explanation of methods and results. Unless this is a requirement for EMBO, it would be better to shorten them.

Figure 1: Quantification of frequency of Y-X NH synapsis in normal (not synchronized) late pachynema cells is lacking.

Figures 3 & 4: data is focused on fl/fl vs cKO cells with full Y-X NH synapsis. It would be interesting to also include data for the remaining cKO spermatocytes.

Figures 6C-d and 6D-d: The way this data is presented is confusing. The percentages do not make sense as one would expect the total number of cells to be scored only for one category. The way this is presented suggests that an X-Y pair lacking signal from all telomeres scores in all three categories. It would be better presented with each spermatocyte scoring for the "most severe" category.

Referee #2:

This research group has made very significant contributions to the understanding of mechanisms related to germ cell development, mammalian meiosis and gametogenesis.

The study does contain multiple interesting findings and not a single key finding. It is of good significance for the field of spermatogenesis and meiotic recombination, and of general interest for the molecular biology community.

This study follows previous published work about the role of Speedy A - CDK2 interaction in the attachment of the telomeres to the nuclear envelope during mouse meiosis. Using an elegant approach of conditional knock out and cell synchronization, here the authors uncover the role of Speedy A and TRF1 in allowing the desynapsis of the X and Y chromosomes in males, and identify a residue in the SUN1 protein which phosphorylation by Speedy A /CDK2 is key for driving chromosomal movement and X-Y desynapsis, necessary for meiotic progression.

Overall, I consider the manuscript to be well written, the reported findings are novel and quite interesting, well presented (figures are clear) and convincing. The conclusions are well supported by the data and are appropriately discussed in the context of earlier literature.

Here is a detailed list of comments, mostly comments of minor importance.

Line 80: It isn't that clear whether or not the full Y chromosome axis is NH synapsed with the X. It has been argued from electron microscopy analyses that NH synapsis goes up to the Y centromere leaving the very tiny Yp short arm unsynapsed. I am not suggesting to change the name of "full X-Y NH synapsis" to "almost full X-Y NY synapsis" all along the manuscript but simply adding some nuance here in the introduction, unless the authors can provide the evidence that Yp is also synapsed.

Line 135: I am not aware of any protein that is exclusively bound on the asynapsed axes of the sex chromosomes in pachynema. It seems that all proteins localized here are also loaded to some extent on late synapsing autosomes (visible in late zygonema - early pachynema transition) or more evidently on asynapsed autosomes in mutant or hybrid contexts. If the authors have found that SpdyA can also be bound to late synapsing autosomes then they should mention it to avoid giving the false impression that SpdyA axis-binding is specific to the sex chromosomes.

Line 211: MLH1 foci count between SpdyA fl/fl and SpdyA cKO with full NH synapsis are comparable, yes, but the distributions on the scatter plot and the averages are really not identical. I think it is important to run a statistical test here and try to discuss this small difference.

Line 218: "the full Y-X NH synapsis appeared independent from meiotic recombination". This needs to be removed or rephrased because this statement is incorrect. If the full and persistent X-Y NH synapsis doesn't affect CO formation at the PAR, one can't state that this synaptic phenomenon is "independent from meiotic recombination". DSB formation at the PAR is part of the "meiotic recombination" process and it is extremely unlikely that the full X-Y NH synapsis (transient in the WT, persistent in the mutant) would still occur in absence of PAR DSBs. Indeed, available data indicate that full X-Y NH synapsis does not occur at all in absence of DSB or even in absence of PAR DSB more specifically.

Line 239: It would be very important to know if desynapsis occurs normally on autosomes during diplotene in the SpdyA mutant. I am not sure if there are enough diplotene cells in the SpdyA cKO to test this, but it seems that there are enough at least in the hypomorphic mutant SpdyA A125V (as shown in Fig. S8G).

Line 285: I think it is more appropriate to plot X axis and Y axis separately as opposed to a Y/X axis ratio, or in addition to the Y-X axis ratio. Informations about potential X chromosome axis length perturbation in the context of full X-Y NH synapsis are also important. One would expect X axis to partly fail to elongate at the limited region of full X-Y NH synapsis, and this information will not appear if only a ratio is plotted.

Line 291: The definition of "irregular loops" as "Y chromatin loops that exhibited an irregular shape, which protruded and dispersed away from the unelongated Y axis" lacks precision to a point where the reader can hardly imagine how the authors used this definition to establish a clean cutoff and find that 79.1% of fully synapsed Y-X in Sync-SpdyA cKO have "irregular loops". Moreover, the following figure, Fig. 4C actually makes the same point as Fig. 4B but with good precision because regions of interest are actually measured. I suggest to remove the Fig. 4B from the manuscript, to ensure clarity and avoid redundancy.

Line 348: "insufficient X-Y chromosome compaction within the sex body, which can be a reason for the cell death." I think the authors should argue for this. There is room in the manuscript for more discussion about what causes cell death in the SpdyA cKO at late pachynema. MSCI appears unaffected. There is no obvious DNA repair defect triggering a checkpoint. An X-Y non-disjunction could lead to apoptosis but only in metaphase as far as I know. So why is cell death occurring at late pachynema and how could it be related to X-Y compaction within the sex body, if it is?

Line 411: It should be noted here that SpdyA signal is also lost from the X axis (not only telomeres) as the Fig 6C shows.

Line 448: Maybe I missed something but I am confused as to why it is "surprising" to find normal tel-NE attachment in the SpdyA A125V given that the tel-NE attachment is also normal in the SpdyA cKO. Is it because the SpdyA A125V mutation is not conditional?

Line 453: The reader will be surprised to find in Fig S6 about the mutation in SPDYA from a patient with non-obstructive azoospermia without being warned in the main text.

Line 455: There is no reference to the Fig. 7A in the main text.

Line 479: I believe "validated with another 3 means" should be replaced by "validated by three other means"

Line 481: It seems that p-SUN(S48) staining shows signal at the sex body that is not visible with the SUN1 antibody. I think the authors should make a comment about this. More confusing however is the fact that this sex body staining appears on the Fig 7E, Fig S9 Ba and Ca, but not on S9 Aa. Some clarification is needed.

Line 856: Fig. 5 legend. What do the white arrowheads indicate?

Line 1078: Fig. S10 title. Replace functional by functionally.

Referee #1

Reviewer comments: This paper by Liu et al extends our understanding of the role of Speedy A in male mouse meiosis. Apart from its previously characterized function in mediating telomere attachment to the nuclear envelope during leptotema, the authors have identified a later-stage requirement for Speedy A for the desynapsis of the non-homologous regions of chromosomes X and Y during pachynema of male meiosis. They carefully dissect this role, showing that the failed desynapsis of the X-Y non-homologous region does not affect MSCI or crossover formation, but instead disrupts the chromatin loop shortening that normally occurs on the Y chromosome at late pachynema. They continue to show that this role of Speedy A is mediated by its phosphorylating (via CDK2) SUN1 protein at S48, revealing a new role for the protein. This also adds new insight for the role of Y-X NH desynapsis during male mouse meiosis. Overall, the study is carried out carefully, with attention to detail and many important findings. Plenty of overlapping evidence to strengthen points is given, and the manuscript's message is clear. However, there are some concerns that need to be addressed before publication.

Authors' response: We sincerely thank the reviewer for his/her positive comments and appreciation of our work. We have carefully addressed all of the reviewer's concerns point by point, and have revised the manuscript accordingly, as listed below.

Reviewer comments-Major 1: What is the fertility status of male mice with conditional deletion of Speedy A? Please comment on this in the text. Could you collect sperm from a later timepoint to perform IVF experiments?

Authors' response: We thank the reviewer for this important question. Our additional experiments showed that the epididymides from tamoxifen-treated *Spdya^{fl/fl}; Ddx4-Cre^{ERT2}* mice lacked spermatozoa, which may lead to infertility. This data has been included as Appendix Fig. S4, and a corresponding paragraph has been added to the main text (Line 278-283):

“As expected, *Sync-Spdya^{CKO}* testes collected one month later (RA45, i.e., PD54) were smaller in size than the control *Sync-Spdya^{fl/fl}* testes and exhibited meiotic arrest at zygonema in the majority of seminiferous tubules, with a marked reduction in tubules containing spermatids (Appendix Fig. S4, A-C). Consequently, few spermatozoa were detected in the epididymides of adult *Sync-Spdya^{CKO}* mice (Appendix Fig. S4D), which may result in infertility.”

Appendix Figure S4.

(A) Synchronized mice treated with tamoxifen were allowed to grow to adulthood, and testes and epididymides were collected at RA45. (B) The *Sync-Spdya^{cKO}* testis was smaller in size than the control *Sync-Spdya^{fl/fl}* testis. Scale bar, 5 mm. (C) Histological analysis revealed meiotic arrest at the zygotene stage in majority of seminiferous tubules in adult *Sync-Spdya^{cKO}* testes. Scale bars, 50 μ m. (D) Compared to *Sync-Spdya^{fl/fl}* mice, *Sync-Spdya^{cKO}* caudal epididymides collected at RA45 lacked spermatozoa. Scale bars, 50 μ m.

Reviewer comments-Major 2: To evaluate the state of MSCI (establishment of which is based on the asynapsed nature of the X and Y chromosomes) the authors performed RNAseq at their PD20 model of synchronized spermatogenesis. However, there are several concerns regarding this analysis: **a.** first, it's unclear whether correlation coefficients are performed on all genes or the most variable ones.

Authors' response: Correlation coefficients were calculated for all genes in the *Sync-Spdya^{fl/fl}* and *Sync-Spdya^{cKO}* groups. We have included this detail in the figure legend of Appendix Fig. S3 (Appendix):

“The correlation coefficients were calculated based on reads counts from all the genes in the *Sync-Spdya^{fl/fl}* and *Sync-Spdya^{cKO}* groups.”

b. the number of total genes is very low for a mouse bulk RNAseq experiment (11,636), compared to an expected number of ~20000.

Authors' response: For transcriptional profile comparisons, we only include protein-coding genes with an average FPKM > 1 in both groups. The numbers of autosomal protein-coding genes (11,636) and sex-chromosomal protein-coding genes (484) are comparable to those reported in our previous studies (Li et al., *PNAS*, 2021; Lin et al., *Cell Reports*, 2024), and other papers (Han et al, 2021; Sangrithi et al, 2017).

c. what were the 34 DE genes identified in the analysis? Is there anything interesting about them?

Authors' response: In our bulk RNA-seq analyses, we identified 34 differentially expressed (DE) genes ($P_{\text{adj}} < 0.05$), including 27 down-regulated and 7 up-regulated protein-coding genes in the *Sync-Spdy*^{CKO} testes when compared to *Sync-Spdy*^{fl/fl} testes. According to literatures, majority of these genes were involved in spermiogenesis, such as *Ccdc42*, *Slc6b1*, *Efcab5*. This result implies that knockout of *Spdy* may affect later spermiogenesis by dysregulating genes required for sperm development. It would be interesting to investigate this hypothesis in future study. For reference, these genes have been listed in dataset EV1 and mentioned in the text (Line 231-234):

“The transcriptional profiles of the two groups were highly similar (Appendix Fig. S3A), with no differentially expressed (DE) protein coding genes on sex chromosomes (0 out of 484) (Fig. 2E) and only 34 DE genes on autosomes (Dataset EV1, Appendix Fig. S3B).”

d. *the comparison is made on bulk testis tissue, so includes all somatic cells of the testis, which express normal levels of X/Y transcripts. This makes interpretation about sex-gene expression in germ cells difficult to extract. The tamoxifen/Cre-mediated mutation, despite being quite efficient, can be quite variable (reported 68.1% in not-synchronized spermatocytes, but 92.3% in synchronized PD20 spermatocytes). Given that this might be different in each tamoxifen injection event, one cannot be certain as to the percentage of cells that are actually mutant in this experiment. To make the claim that MSCI is normal, single-cell RNAseq could be performed to ensure that only mutated cells are taken into account (judged by the lack of Speedy A transcript). Alternatively, RNA FISH for X and Y transcripts can be performed on chromosome spreads in conjunction with Speedy A IF, to only measure mutated cells. If the authors cannot perform this experiment, they should try reducing their expression fold-change cut-offs to a lower value (e.g. log2FC=0.5), and reassess expression.*

Authors' response: We appreciate the reviewer's very constructive comments. Following his/her suggestions, we performed single-cell RNA-seq analysis on *Sync-Spdy*^{fl/fl} and *Sync-Spdy*^{CKO} mid-late pachynema testis samples, analyzing only spermatocytes. With a log2FC cutoff of 0.58, there were 30 autosomal genes differentially expressed in *Sync-Spdy*^{CKO} spermatocytes, and no sex-chromosomal genes were differentially expressed. These findings confirm normal MSCI in *Sync-Spdy*^{CKO} spermatocytes. We have included these single-cell RNA-seq data in the revised manuscript (Line 234-247):

“To focus on spermatocyte transcriptomes, we further performed single-cell RNA sequencing (scRNA-seq) analyses with *Sync-Spdy*^{fl/fl} and *Sync-Spdy*^{CKO} testes at RA11.75 (Fig. EV2). After filtering out low-quality cells, a total of 19,708 *Sync-Spdy*^{fl/fl} and 19,767 *Sync-Spdy*^{CKO} testicular cells were retained for analysis. Uniform manifold approximation and projection (UMAP) analysis showed unsupervised clustering of these cells, and 3 spermatogenic and 7 somatic cell types were identified by established cell-specific markers (Fig. EV2A) (Ernst *et al*, 2019; Green *et al*, 2018). Marker gene expression differences among these main cell types were visualized with a heatmap (Fig. EV2B). Compared to *Sync-Spdy*^{fl/fl} testes, *Sync-Spdy*^{CKO} testes showed a lower percentage of spermatocytes and a higher percentage of spermatogonia, indicative of spermatocyte loss (Fig. EV2C). Finally, 3623 *Sync-*

Spdya^{fl/fl} and 2187 *Sync-Spdya*^{CKO} spermatocytes were subjected to DE gene analysis, and 22 autosomal genes (Dataset EV2) but no sex-chromosome genes were identified as DE genes (Fig. EV2D).”

Figure EV2.

(A) UMAP and clustering analysis of single-cell transcriptome data from testicular cells collected at RA11.75 from tamoxifen-treated *Spdya*^{fl/fl} and *Spdya*^{fl/fl}, *Ddx4-Cre*^{ERT2} mice. (B) Heatmap showing expression of marker genes for 10 identified cell types (right). (C) Proportion of the three types of spermatogenic cell types (right). (D) Differential expression analysis of 2,187 *Sync-Spdya*^{CKO} and 3,623 *Sync-Spdya*^{fl/fl} pachytene spermatocytes.

Reviewer comments-Major 3. Cell death is measured by c-PARP staining on testis sections. This is only commented as "many c-PARP positive pachytene cells" (line 268), but no quantification is provided.

Authors' response: We appreciate the reviewer's comments and have included the quantification data for c-PARP positive pachytene cells in the revised text (Line 315-318):

"Immunostaining of late pachytene *Sync-Spdya*^{CKO} testis sections with γ H2AX and c-PARP (cleaved poly ADP ribose polymerase), an apoptosis marker, revealed a significant increase in c-PARP-positive pachytene cells (Fig. EV3D-b, arrows), indicating apoptosis of *Sync-Spdya*^{CKO} spermatocytes at late pachynema."

Figure EV3D

Following Regimen B, late pachytene *Sync-SpdyA*^{CKO} spermatocytes collected at RA13 exhibited apoptosis. Testis sections were immunostained for c-PARP (green) and γH2AX (red). Nuclei were counterstained with DAPI. Scale bars, 30 μm. “n” represents the total number of seminiferous tubules scored from three mice per genotype.

-Does deletion of Speedy A obstruct progression to diplonema and further?

Authors' response: We thank the reviewer for this insightful question. Under the tamoxifen regimen illustrated in Figs. 2 and 3, pachytene spermatocytes were eliminated in *Sync-SpdyA*^{CKO} testes following prolonged depletion of SpdyA, thereby impeding meiotic progression to diplonema. This obstruction is attributed to severe telomere-nuclear envelope (Tel-NE) detachment observed in mid-pachytene *Sync-SpdyA*^{CKO} spermatocytes. To clarify this point, we have included a new subsection (Result 5) in the revised text (Line 264-276):

“Continuous SpdyA function is required to maintain Tel-NE attachment and ensure spermatocyte survival throughout pachynema.

We found that 47.4% of mid-pachytene *Sync-SpdyA*^{CKO} cells (RA11.25) exhibited telomere (Tel)-NE detachment, with at least one TRF1 focus observed within the nuclear interior (Fig. EV3A-b, arrows, and c). In contrast, only 11.6 % of *Sync-SpdyA*^{fl/fl} cells showed Tel-NE detachment (Fig. EV3A-c). Consequently, *Sync-SpdyA*^{CKO} testes (RA13.25), in which *SpdyA* was ablated for an extended period, were smaller in size and lacked late pachytene spermatocytes (Fig. EV3B, a and c, asterisk; Fig. EV3C, Regimen A). By comparison, the control *Sync-SpdyA*^{fl/fl} testes contained late pachytene spermatocytes (Fig. EV3B-b). These results suggest that SpdyA is not only essential for initiating the Tel-NE tethering during zygonema, but is also critical for maintaining Tel-NE attachment and supporting spermatocyte survival throughout pachynema”.

Figure EV3A-C.

(A) Compared to *Sync-Spdya^{fl/fl}* cells (a), mid-pachytene (RA11.25) *Sync-Spdya^{cKO}* cell (b) exhibited severe Tel-NE detachment (arrows). (c) Percentages of pachytene cells with Tel-NE detachment. (B) Following tamoxifen regimen shown in Fig. 2A, the *Sync-Spdya^{cKO}* testes collected at RA13.25 were smaller in size and lacked late pachytene spermatocytes. LP, late pachytene spermatocyte; asterisk indicates loss of spermatocytes. Scale bars, 30 μ m. (C) Summary of tamoxifen treatment regimens used to obtain mid- and late pachytene *Sync-Spdya^{cKO}* spermatocytes.

-Are any spermatids observed at later timepoints? Are these (if any) spermatids aneuploid?

Authors' response: As all *Spdya^{cKO}* spermatocytes were eliminated following late pachynema, no spermatids were derived from them. The spermatids observed in Appendix Fig. S4 may be derived from unaffected spermatogonia.

Reviewer comments-Major 4. When harvesting testes at PD20 to look at mid-pachynema, the authors inject tamoxifen at PD16/17, which in their synchronized model of spermatogenesis corresponds to leptozygonema. However, to look at late pachynema (harvesting at PD22) they inject tamoxifen at days PD18/19, which corresponds to late zygonema/early pachynema. Therefore, mutation of the gene happens at different points during meiotic progression and can affect the results. All experiments performed at PD22 should be consistent with the PD20 experiments and should involve deletion at a similar timeframe. This is relevant because conclusions are drawn regarding the continuity of the process based on data from both timepoints. For example, to provide evidence that late-pachynema spermatocytes do not exhibit desynapsis (and therefore the persistence of the Y-X NH synapsis lasts from early to late pachynema), they carry out quantification of synapsed X-Y NH regions at PD22. However, this does not prove that the spermatocytes observed at PD20 would behave the same if left for 2 more days without the protein, and more importantly it does not prove that PD20 spermatocytes are not delayed and would not resolve the Y-X NH

synapsis by PD22. Similarly, to look at early pachynema (PD19 collection), tamoxifen injections are carried out a day earlier than those for PD20 collections. It's unclear to us why these different timings were used; the authors should explain.

Authors' response: We sincerely appreciate the reviewer's careful and insightful analysis of our data and related comments, which have tremendously helped us improve the presentation of our study. Our experiments utilize an inducible knockout mouse model that allows for gene deletion at specific developmental stages. In general, this approach is particularly advantageous when studying genes whose knockout results in early cell death, thereby hindering the investigation of gene functions in later stages. By delaying gene deletion by 1–2 days, we can induce the same phenotype at a later time point, facilitating the study of gene functions during late-stage cell development.

With tamoxifen injection at PD16 (Fig. 2), we were unable to obtain late pachytene *Spdya*^{CKO} spermatocytes at PD22 (i.e., RA13), because the *Spdya*^{CKO} cells were eliminated before reaching late pachynema at PD22. In order to study details of the full *Y-X NH synapsis* in late pachytene *Spdya*^{CKO} spermatocytes, we employed a delayed tamoxifen injection regimen at PD18 (Fig. 3A). Thus, as summarized Fig. EV3C (Regimen A), we utilized mid-late pachytene spermatocytes at PD20, derived from tamoxifen induction at PD16 (Fig. 2A), to examine several meiotic events, such as H1t replacement, crossover formation, MSCI and sex body formation. Then, we use late pachytene spermatocytes at PD22, derived from tamoxifen induction at PD18 (Fig. EV3C, Regimen B), to study X-Y desynapsis and X-Y chromosome compaction.

To clarify the purpose for different tamoxifen regimens, we have incorporated the below illustration into Fig. EV3 with detailed figure legend (Line 1386-1389):

“(C) Summary of tamoxifen treatment regimens used to obtain mid- and late pachytene *Sync-Spdya*^{CKO} spermatocytes. Regimens A and B correspond to those shown in Figs. 2A and 3A, respectively. Dashed lines with crosses indicate stages at which *Sync-Spdya*^{CKO} spermatocytes could not be obtained.”

Figure EV3C.

Reviewer comments-Major 5. Given that in this mutant the X and Y undergo persistent synapsis, and that part of the Y doesn't accumulate MSCI-associated repressive marks, it is puzzling that the Y doesn't escape MSCI. The authors should address this in their discussion.

Authors' response: We appreciate the reviewer's valuable suggestion. We have added a paragraph to address this issue in the revised discussion (Line 600-608):

“One intriguing finding in our study is that the fully NH synapsed X-Y chromosomes do not escape MSCI. Why is this the case? X-Y synapsis is highly regulated and occurs specifically at early pachynema, a stage when MSCI has already been initiated across the X-Y chromosomes. MSCI is typically established by mid-pachynema and is known to remain remarkably stable once established (Royo *et al*, 2013). It is possible that from early to mid-pachynema, the loop-axis reorganization of sex chromosomes is not as pronounced as it is during the late pachytene stage, thereby allowing MSCI to be properly established in *Spdya*^{cKO} pachytene spermatocytes. This hypothesis remains to be tested in future studies.”

Reviewer comments-Minor 1:

Figure legends are long and contain extensive explanation of methods and results. Unless this is a requirement for EMBO, it would be better to shorten them.

Authors' response: We have shortened the figure legends in the revised manuscript.

Reviewer comments-Minor 2:

Figure 1: Quantification of frequency of Y-X NH synapsis in normal (not synchronized) late pachynema cells is lacking.

Authors' response: We have included this result in the revised text (Line 164-166):

“Quantitative analysis revealed that 47.1% of *Spdya*^{cKO} pachytene cells exhibited *full Y-X NH synapsis*, compared to only 3.3% of *Spdya*^{fl/fl} cells (Fig. 1C-c).”

Figure 1C-c.

Percentages of pachytene cells with different X-Y configurations in *Spdya*^{fl/fl} and *Spdya*^{cKO} testes.

Reviewer comments-Minor 3:

Figures 3 & 4: data is focused on fl/fl vs cKO cells with full Y-X NH synapsis. It would be interesting to also include data for the remaining cKO spermatocytes.

Authors' response: We appreciate the reviewer's insightful suggestion. Our data indicate that the end-to-end associated X-Y chromosomes in *Spdya*^{cKO} spermatocytes

exhibit normal loop-axis organization. This finding further supports the correlation between *full Y-X NH synapsis* and disrupted loop-axis organization. We have incorporated these data into the revised text (Line 372-374):

“Similar to *Sync-SpdyA^{fl/fl}* cells, late pachytene *Sync-SpdyA^{CKO}* cells with end-to-end associated X-Y chromosomes showed compact Y chromatin loops enclosed within aggregated γ H2AX or SCML2 areas (Appendix Fig. S6A and S6B).”

Appendix Figure S6A-B.

(A) Immunofluorescence analysis of late pachytene *Sync-SpdyA^{CKO}* spermatocytes with end-to-end X-Y configuration, stained for γ H2AX (light blue), Chr Y (green) and SYCP3 (red). Scale bar, 2 μ m. (B) Immunofluorescence analysis with SCML2 (light blue), Chr Y (green) and SYCP3 (red). Scale bar, 2 μ m.

Reviewer comments-Minor 4:

Figures 6C-d and 6D-d: The way this data is presented is confusing. The percentages do not make sense as one would expect the total number of cells to be scored only for one category. The way this is presented suggests that an X-Y pair lacking signal from all telomeres scores in all three categories. It would be better presented with each spermatocyte scoring for the "most severe" category.

Authors' response: We apologize for the confusion. We have followed the reviewer's suggestion and present these bar graphs in a new way (Line 457-460):

“Quantitative analysis revealed that among the *Sync-Trf1^{CKO}* pachytene cells with *full Y-X NH synapsis*, 42.0% lacked SpdyA specifically at the non-PAR telomeres of the Y chromosome, 51.7% lacked SpdyA at the non-PAR telomeres of both X and Y chromosomes, and 6.3% lacked SpdyA at all telomeres (Fig. 6C-d).”

Figure 6C-d

Percentages of X-Y pairs with SpdyA signal lost from different telomeres in *Sync-Trf1^{ckO}* pachytene cells with full Y-X NH synapsis.

And Line 474-477:

“Quantification showed that 49.1% of the *Sync-Trf1^{ckO}* pachytene cells with full Y-X NH synapsis lacked SUN1 at non-PAR telomeres of the Y chromosome, 39.7% lacked SUN1 at non-PAR telomeres of both X and Y chromosomes, and 11.2% lacked SUN1 at all telomeres (Fig. 6D-d).”

Figure 6D-d

Percentages of X-Y pairs with SUN1 signal lost from different telomeres in *Sync-Trf1^{ckO}* pachytene cells with full Y-X NH synapsis.

References

1. Li M, Zheng J, Li G, Lin Z, Li D, Liu D, et al. The male germline-specific protein MAPS is indispensable for pachynema progression and fertility. *Proceedings of the National Academy of Sciences of the United States of America* 2021, 118(8).
2. Lin Z, Li D, Zheng J, Yao C, Liu D, Zhang H, et al. The male pachynema-specific protein MAPS drives phase separation in vitro and regulates sex body formation and chromatin behaviors in vivo. *Cell reports* 2024, 43(1): 113651.
3. Han G, Hong SH, Lee SJ, Hong SP, Cho C (2021) Transcriptome Analysis of Testicular Aging in Mice. *Cells* 10
4. Sangrithi MN, Royo H, Mahadevaiah SK, Ojarikre O, Bhaw L, Sesay A, Peters AH, Stadler M, Turner JM (2017) Non-Canonical and Sexually Dimorphic X Dosage Compensation States in the Mouse and Human Germline. *Developmental cell* 40: 289-301.e283
5. Ernst C, Eling N, Martinez-Jimenez CP, Marioni JC, Odom DT (2019) Staged developmental mapping and X chromosome transcriptional dynamics during mouse spermatogenesis. *Nature communications* 10: 1251

6. Green CD, Ma Q, Manske GL, Shami AN, Zheng X, Marini S, Moritz L, Sultan C, Gurczynski SJ, Moore BB *et al* (2018) A Comprehensive Roadmap of Murine Spermatogenesis Defined by Single-Cell RNA-Seq. *Developmental cell* 46: 651-667.e610.
7. Royo H, Prosser H, Ruzankina Y, Mahadevaiah SK, Cloutier JM, Baumann M, Fukuda T, Höög C, Tóth A, de Rooij DG *et al* (2013) ATR acts stage specifically to regulate multiple aspects of mammalian meiotic silencing. *Genes & development* 27: 1484-1494.

Referee #2

Reviewer comments: *This research group has made very significant contributions to the understanding of mechanisms related to germ cell development, mammalian meiosis and gametogenesis.*

The study does contain multiple interesting findings and not a single key finding. It is of good significance for the field of spermatogenesis and meiotic recombination, and of general interest for the molecular biology community.

This study follows previous published work about the role of Speedy A - CDK2 interaction in the attachment of the telomeres to the nuclear envelope during mouse meiosis. Using an elegant approach of conditional knock out and cell synchronization, here the authors uncover the role of Speedy A and TRF1 in allowing the desynapsis of the X and Y chromosomes in males, and identify a residue in the SUN1 protein which phosphorylation by Speedy A /CDK2 is key for driving chromosomal movement and X-Y desynapsis, necessary for meiotic progression. Overall, I consider the manuscript to be well written, the reported findings are novel and quite interesting, well presented (figures are clear) and convincing. The conclusions are well supported by the data and are appropriately discussed in the context of earlier literature.

Authors' response: We sincerely thank the reviewer for his/her thoughtful and encouraging comments. We are grateful that the reviewer recognizes the novelty and significance of our findings, as well as the clarity of our data presentation and the strength of our conclusions. We also appreciate the reviewer's acknowledgement of our group's continued efforts in dissecting the molecular mechanisms underlying germ cell development and meiosis. We have revised the manuscript carefully to address all suggestions and comments from the reviewer, and further improve the clarity and impact of our study. The revision is listed point-by-point below.

Reviewer comments-1: *Here is a detailed list of comments, mostly comments of minor importance.*

Line 80: It isn't that clear whether or not the full Y chromosome axis is NH synapsed with the X. It has been argued from electron microscopy analyses that NH synapsis goes up to the Y centromere leaving the very tiny Yp short arm unsynapsed. I am not suggesting to change the name of "full X-Y NH synapsis" to "almost full X-Y NH synapsis" all along the manuscript but simply adding some nuance here in the

introduction, unless the authors can provide the evidence that Yp is also synapsed.

Authors' response:

We thank the reviewer for this insightful comment. As shown in the new Fig. 5C (also referenced below), continuous SYCP1 signals span the entire, or nearly entire Y chromosome axis, including through the Y centromere, in early pachytene spermatocytes. Similar patterns were also observed in Figs. 1D-c, 6B-c. Therefore, our evidence supports that Yp is also synapsed during early pachynema in mouse spermatocytes. In response to the reviewer's suggestion, we have revised the corresponding sentence in the introduction (Line 78-81) to include more nuanced language regarding this point:

“As illustrated in Fig. EV1, during pachynema in mouse spermatocytes, the X and Y chromosomes first undergo a non-homologous (NH) synapsis with synaptonemal complex (SC) built between the entire (or nearly entire) Y axis and the X axis (Fig. EV1A-a).”

Figure for rebuttal letter

White arrow indicates SC built between X and Y chromosomes. Yp, short arm of Y chromosome (red arrows), which is SYCP1-positive.

Reviewer comments-2: Line 135: *I am not aware of any protein that is exclusively bound on the asynapsed axes of the sex chromosomes in pachynema. It seems that all proteins localized here are also loaded to some extent on late synapsing autosomes (visible in late zygonema - early pachynema transition) or more evidently on asynapsed autosomes in mutant or hybrid contexts. If the authors have found that SpdyA can also be bound to late synapsing autosomes then they should mention it to avoid giving the false impression that SpdyA axis-binding is specific to the sex chromosomes.*

Authors' response: We concur with the reviewer's comments and have confirmed the localization of SpdyA at the unsynapsed regions of late-synapsing autosomes in wild-type late zygotene spermatocytes. This result has been added as a subfigure (Fig. EV1C) in the revised manuscript and is described in the main text (Line 112-114):

“Additionally, axial localization of SpdyA is also observed at the unsynapsed region of late-synapsing autosomes in late zygotene spermatocytes (Fig. EV1C).” highlighted in yellow.

Figure EV1C

SpdyA localization in the unsynapsed region (dashed circle) of late-synapsing autosomes at late zygonema. Scale bar, 2 μ m.

Reviewer comments-3: Line 211: *MLH1 foci count between SpdyA fl/fl and SpdyA cKO with full NH synapsis are comparable, yes, but the distributions on the scatter plot and the averages are really not identical. I think it is important to run a statistical test here and try to discuss this small difference.*

Authors' response: We agree with the reviewer and have run a statistical test which shows modest but significant difference between these two groups. This statistical result has been added to Fig. 2D-c and legend, and is discussed in the main text (Line 218-225):

“Compared to *Sync-SpdyA^{fl/fl}* pachytene cells, *Sync-SpdyA^{cKO}* cells showed a modest yet significant reduction in total MLH1 focus counts (Fig. 2D, arrows in a and b; Fig. 2D-c). However, 72.6% of the *Sync-SpdyA^{cKO}* cells with *full Y-X NH synapsis* exhibited one MLH1 focus in the PAR of the sex chromosomes versus 60.6% in the *Sync-SpdyA^{fl/fl}* control cells (Fig. 2D, arrowheads in insets of a and b; and Fig. 2D-d). These results indicate that the *full Y-X NH synapsis* in *Sync-SpdyA^{cKO}* pachytene spermatocytes is unlikely due to retarded pachynema progression. It seems that crossover formation in the PAR was not interrupted in *Sync-SpdyA^{cKO}* pachytene cells, and that Y-X NH desynapsis appeared independent from meiotic recombination.”

Figure 2D-c.

Scatter plot showing a modest reduction in total MLH1 focus counts per cell in *Sync-SpdyA^{cKO}* spermatocytes (closed dots) compared to *Sync-SpdyA^{fl/fl}* spermatocytes (open dots).

Reviewer comments-4: Line 218: *"the full Y-X NH synapsis appeared independent from meiotic recombination". This needs to be removed or rephrased because this statement is incorrect. If the full and persistent X-Y NH synapsis doesn't affect CO formation at the PAR, one can't state that this synaptic phenomenon is "independent*

from meiotic recombination". DSB formation at the PAR is part of the "meiotic recombination" process and it is extremely unlikely that the full X-Y NH synapsis (transient in the WT, persistent in the mutant) would still occur in absence of PAR DSBs. Indeed, available data indicate that full X-Y NH synapsis does not occur at all in absence of DSB or even in absence of PAR DSB more specifically.

Authors' response: We appreciate the reviewer's attention to detail. Indeed, the *full X-Y NH synapsis* is initiated by DSB-promoted X-Y synapsis in the PAR at late zygonema. This statement has been corrected in the text (Line 225-227):

“It seems that crossover formation in the PAR was not interrupted in *Sync-SpdyA^{cKO}* pachytene cells, and that Y-X NH desynapsis appeared independent from meiotic recombination.”

Reviewer comments-5: *Line 239: It would be very important to know if desynapsis occurs normally on autosomes during diplotene in the SpdyA mutant. I am not sure if there are enough diplotene cells in the SpdyA cKO to test this, but it seems that there are enough at least in the hypomorphic mutant SpdyA A125V (as shown in Fig. S8G).*

Authors' response: we thank the reviewer for this insightful suggestion. To assess whether autosomal desynapsis occurs normally in diplotene *SpdyA^{cKO}* spermatocytes, we performed additional experiments. Because SpdyA depletion leads to pachytene spermatocyte death (Fig. EV3), we were unable to obtain diplotene *SpdyA^{cKO}* spermatocytes using tamoxifen regimen A or B (see figure below). Therefore, we employed regimen C (see figure below), in which 2 consecutive *i.p.* tamoxifen injections (20 mg/kg body weight) were administered at RA10 and RA11 (i.e., PD19 and PD20), and testicular cells were collected at RA14.25 (i.e., PD23). Based on SYCP3, SYCP1 and SpdyA triple staining, we observed normal autosomal desynapsis in diplotene *SpdyA^{cKO}* spermatocytes (see figure below). These findings suggest that SpdyA is dispensable for autosomal desynapsis. We have included these data and revised the text (Line 326-327):

“In contrast, our results indicate that SpdyA is dispensable for autosomal desynapsis (Appendix Fig. S5).”

Appendix Figure S5.

(A) Tamoxifen treatment regimen for generating diplotene *Sync-SpdyA^{CKO}* spermatocytes. (B) Chromosome spreads of diplotene *Sync-SpdyA^{fl/fl}* and *Sync-SpdyA^{CKO}* spermatocytes. Scale bars, 5 μ m. (c) Quantification desynapsed autosomes per diplotene spermatocyte in both genotypes. Data are presented as mean \pm SD; ns, not significant (Mann–Whitney test).

Reviewer comments-6: Line 285: I think it is more appropriate to plot X axis and Y axis separately as opposed to a Y/X axis ratio, or in addition to the Y-X axis ratio. Informations about potential X chromosome axis length perturbation in the context of full X-Y NH synapsis are also important. One would expect X axis to partly fail to elongate at the limited region of full X-Y NH synapsis, and this information will not appear if only a ratio is plotted.

Authors' response: We agree with the reviewer's comment and have now presented the lengths of the X and Y axes separately in the revised text (Line 333-340):

“By measuring the lengths of the X and Y axes (Fig. 4A, a and b), we found that while X axis length was only modestly shortened (Fig. 4A-c), both the Y axis length and the Y/X axis ratio were significantly reduced in late pachytene *Sync-SpdyA^{CKO}* cells with full Y-X NH synapsis, compared to *Sync-SpdyA^{fl/fl}* cells (Fig. 4A, d and e). These results indicate that full Y-X NH synapsis may perturb the lengthening of the entire Y axis, and to a lesser extent the X axis, in *Sync-SpdyA^{CKO}* cells during late pachynema, whereas X-Y axis lengthening proceeds normally in *Sync-SpdyA^{fl/fl}* cells.”

Figure 4A.

X-Y axis length measurements in late pachytene (RA13) *Sync-Spdya*^{fl/fl} (a) and *Sync-Spdya*^{cKO} (b) spermatocytes. Green and blue dashed lines show X and Y chromosome axes, respectively. (c-e) Measured lengths of the X axis (c), Y axis (d), and the calculated Y/X axis length ratio (e) in late pachytene *Sync-Spdya*^{fl/fl} cells and *Sync-Spdya*^{cKO} cells with full Y-X NH synapsis. Each dot represents a single nucleus.

Reviewer comments-7: Line 291: The definition of "irregular loops" as "Y chromatin loops that exhibited an irregular shape, which protruded and dispersed away from the unelongated Y axis" lacks precision to a point where the reader can hardly imagine how the authors used this definition to establish a clean cutoff and find that 79.1% of fully synapsed Y-X in *Sync-Spdya* cKO have "irregular loops". Moreover, the following figure, Fig. 4C actually makes the same point as Fig. 4B but with good precision because regions of interest are actually measured. I suggest to remove the Fig. 4B from the manuscript, to ensure clarity and avoid redundancy.

Authors' response: We appreciate the reviewer's careful and insightful evaluation of our data, which has very much improved the clarity of our study. In response to the reviewer's suggestion, we have removed the original Fig. 4B from the manuscript to reduce redundancy and enhance overall clarity.

Reviewer comments-8: Line 348: "insufficient X-Y chromosome compaction within the sex body, which can be a reason for the cell death." I think the authors should argue for this. There is room in the manuscript for more discussion about what causes cell death in the *SpdyA* cKO at late pachynema. MSCI appears unaffected. There is no obvious DNA repair defect triggering a checkpoint. An X-Y non-disjunction could lead to apoptosis but only in metaphase as far as I know. So why is cell death occurring at late pachynema and how could it be related to X-Y compaction within the sex body, if it is?

Authors' response: The reviewer has raised a truly important and interesting point by saying "An X-Y non-disjunction could lead to apoptosis but only in metaphase as far as I know. So why is cell death occurring at late pachynema...". Although establishing a direct causal link between X-Y non-disjunction and the death of pachytene spermatocytes has been challenging in the *SpdyA*^{cKO} model because the *SpdyA*^{cKO} pachytene spermatocytes undergo rapid demise, we did utilize the hypomorphic *SpdyA*^{A125V} mouse model to investigate this relationship. In the *SpdyA*^{A125V} mice, over 47% of X-Y chromosome pairs exhibited full Y-X NH synapsis, with normal telomere-nuclear envelope (Tel-NE) attachment, H1t replacement, and

crossover formation. Despite these normal meiotic features, a significant proportion of *Spdya*^{A125V} pachytene spermatocytes still failed to progress to diplotene and underwent cell death. This observation strongly suggests that insufficient X-Y chromosome compaction within the sex body, rather than synapsis or crossover defects, may be one of the reasons driving cell death during pachytene arrest in *Spdya*^{A125V} spermatocytes. In the revised manuscript, we have revised the main text and expanded the discussion on this mechanism, emphasizing the potential role of chromosomal compaction dynamics in meiotic checkpoint surveillance.

In the result (Line 500-509):

“We further found that *Spdya*^{A125V} pachytene spermatocytes exhibited normal Tel-NE attachment (Fig. EV4G), as well as normal H1t replacement and crossover formation (Appendix Fig. S11). However, in late pachytene *Spdya*^{A125V} cells with *full Y-X NH synapsis*, the loop-axis organization of the Y chromosome—and possibly also the X chromosome—was disrupted (Fig. EV4H). This defect was not observed in *Spdya*^{A125V} cells with end-to-end associated sex chromosomes (Appendix Fig. S6C). Based on these findings, we conclude that the disrupted loop-axis organization of sex chromosomes, caused by persistent *full Y-X NH synapsis*, is likely one of the key contributors to the elimination of *Spdya*^{A125V} pachytene cells, and that pachytene spermatocytes are particularly vulnerable to hypomorphic SpdyA function.”

Figure EV4G-H.

(G) IF analyses of *Spdya*^{+/+} and *Spdya*^{A125V} testis sections with Lamin B1 (blue), TRF1 (green) and SYCP1 (red). Arrows indicate Tel-NE attachment. Scale bars, 5 μm. (c) Percentages of *Spdya*^{+/+} and *Spdya*^{A125V} pachytene cells with intact Tel-NE attachment. (H) Sex chromosomes in late pachytene *Spdya*^{A125V} spermatocytes with *full Y-X NH synapsis* displayed disrupted loop-axis organization. Scale bars, 5 μm.

Appendix Figure S6C

Immuno-FISH analysis of late pachytene *Spdya*^{A125V} spermatocytes with end-to-end X-Y configuration, stained for γ H2AX (light blue), Chr Y (green) and SYCP3 (red). Scale bar, 2 μ m.

And in the discussion (Line 610 -623):

“An important unresolved question in our study is the mechanism underlying the death of spermatocytes with persistent *full Y-X NH synapsis*. At present, it remains challenging to establish a direct causal link between disrupted X-Y loop-axis organization and the cell death seen in the *Spdya*^{CKO} mouse model, which exhibits severe Tel-NE detachment that may compromise the nuclear lamina and affect spermatocyte survival (Zetka *et al*, 2020). However, our analysis of *Spdya*^{A125V} mice—where Tel-NE attachment, H1t replacement, and crossover formation are preserved—revealed a strong correlation between *full Y-X NH synapsis* and disrupted X-Y loop-axis organization in late pachytene spermatocytes. We also observed that a big proportion of *Spdya*^{A125V} pachytene spermatocytes failed to progress to diplotema and instead underwent apoptosis. These findings suggest a potential role for chromosomal compaction dynamics in meiotic checkpoint surveillance. We propose that further investigation using the *Spdya*^{A125V} model will help elucidate how impaired X-Y chromosomal compaction contributes to pachytene spermatocyte apoptosis.”

Reviewer comments-9: Line 411: It should be noted here that *SpdyA* signal is also lost from the X axis (not only telomeres) as the Fig 6C shows.

Authors' response: We appreciate the reviewer's careful evaluation of our data. In Fig. 6C, our analysis primarily focused on *SpdyA* localization at the telomeres of sex chromosomes in *Sync-Trf1*^{CKO} pachytene spermatocytes. However, as suggested by the reviewer, we have noted this point in the figure legend (Line 1304-1306):

“A minority of sex chromosomes with *full Y-X NH synapsis* showed a loss of *SpdyA* from all telomeres (c-c', white arrows), as well as from the sex chromosome axes.”

Reviewer comments-10: Line 448: Maybe I missed something but I am confused as to why it is "surprising" to find normal tel-NE attachment in the *SpdyA* A125V given that the tel-NE attachment is also normal in the *SpdyA* cKO. Is it because the *SpdyA* A125V mutation is not conditional?

Authors' response: Based on prior studies showing that *Spdya* knockout disrupts telomere tethering to the nuclear envelope (Tel-NE) in zygotene spermatocytes, we initially hypothesized that *Spdya*^{A125V} mice would display a similar defect. However, no such abnormality was observed in this *Spdya* hypomorphic mouse model. To avoid ambiguity, we have removed the phrase “to our surprise” in the text.

Reviewer comments-11: Line 453: The reader will be surprised to find in Fig S6 about the mutation in *SPDYA* from a patient with non-obstructive azoospermia without being warned in the main text.

Authors' response: We thank the reviewer for this helpful observation. In response, we have revised the main text to introduce the origin of *Spdya*-A125V mutation more

clearly (Line 487-489):

“We generated a hypomorphic-SpdyA mouse model, the *SpdyA*^{A125V} knockin mice, which recapitulate a missense mutation originally identified in a patient with non-obstructive azoospermia (NOA, Appendix Fig. S9)”.

Reviewer comments-12: Line 455: *There is no reference to the Fig. 7A in the main text.*

Authors' response: We have corrected this in the main text (Line 513-517):

“To identify the physiological substrates of SpdyA/CDK2 kinases in mediating Y-X NH desynapsis, we performed phosphoproteomic analyses on testes from *Sync-SpdyA*^{fl/fl} and *Sync-SpdyA*^{CKO} mice, which were collected at RA10.25, which is when over 75% of *Sync-SpdyA*^{CKO} pachytene cells exhibit *full Y-X NH synapsis* (Figs. 7A & 5A).”

Reviewer comments-13: Line 479: *I believe "validated with another 3 means" should be replaced by "validated by three other means"*

Authors' response: We thank the reviewer and have revised this.

Reviewer comments-14: Line 481: *It seems that p-SUN(S48) staining shows signal at the sex body that is not visible with the SUN1 antibody. I think the authors should make a comment about this. More confusing however is the fact that this sex body staining appears on the Fig 7E, Fig S9 Ba and Ca, but not on S9 Aa. Some clarification is needed.*

Authors' response: The signal of the p-SUN1(S48) staining at the sex body area in pachytene cells appeared to be non-specific. This may be due to the use of a short-phosphorylated peptide as the immunogen for antibody production. We have added a sentence to address this point in the main text (Line 540-542):

“The p-SUN1(S48) signals observed in the sex body region were non-specific, as the unphosphorylated SUN1 was exclusively located at telomeres (Fig. 7E; Appendix Fig. S12B).”

Reviewer comments-15: Line 856: *Fig. 5 legend. What do the white arrowheads indicate?*

Authors' response: The white arrowheads indicate Tel-NE attachment. We have revised the legend.

Reviewer comments-16: Line 1078: *Fig. S10 title. Replace functional by functionally.*

Authors' response: We have revised the text.

References

1. Zetka M, Paouneskou D, Jantsch V (2020) "The nuclear envelope, a meiotic jack-

of-all-trades". *Curr Opin Cell Biol* 64: 34-42.

Prof. Kui Liu
The University of Hong Kong
Department of Obstetrics and Gynecology, Li Ka Shing Faculty of Medicine
Hong Kong Special Administrative Region of the People's Republic of China

23rd Jul 2025

Re: EMBOJ-2025-120432R
Speedy A governs non-homologous XY chromosome desynapsis as a unique prerequisite for XY loop-axis organization

Dear Dr. Liu,

Thank you for submitting your revised manuscript to The EMBO Journal. I am happy to say that after re-review by the original referees, who both considered their concerns overall satisfactorily addressed, we have now accepted it for publication!

As you will see below, referee 1 retained some questions regarding the (sc)RNA-seq analyses, which would in my opinion at this stage not require further modifications to the manuscript; but it would be good if you could briefly respond to these points via email with a final response letter.

With kind regards,

Hartmut Vodermaier

Referee #1:

It is evident that the authors have carefully addressed all the comments provided. While most answers are satisfactory, there is one remaining issue with the RNAseq and scRNAseq analysis:

RNAseq: The FPKM > 1 filter that is used here is quite strict. It would not be surprising if most Y transcripts are not picked up at all. Have you tried analyzing the data using DESeq2?

scRNAseq: The generated UMAPs look quite different from other published datasets. In most published datasets, the differentiating spermatogonia and spermatocyte clusters form a continuous trajectory (example of P10 and adult testis dataset DOI: 10.1126/science.ads6495). In addition, the purpose of the scRNAseq is to distinguish cells that are undergoing MSCI in WT samples and to compare that cluster to the mutant. However, the authors have compared the whole spermatocyte populations, which would include cells in leptoneuma (normal X-Y expression), zygoneuma (high X-Y expression) and pachyneuma (low to no expression from early to mid pachyneuma). To make the comparison valid, it should be carried out only between cells in mid-pachyneuma, which would require further clustering of the SC cluster (see for list of marker genes, DOI: 10.1038/s41586-022-05547-7, DOI: 10.1038/s41467-019-09182-1).

The above would make the RNAseq analysis more robust and trustworthy, but it is not essential to the paper as a whole, but rather to the observation that MSCI is unaffected. Whichever the case, the impact of the paper would be similar, and so I would leave it to the discretion of the editor as to whether the manuscript should be accepted in the current format or after these changes are made.

Referee #2:

I just read the authors' revisions and I was perfectly satisfied by their efforts to address the comments point by point. The quality of the manuscript has significantly improved and I believe is ready for publication.

Referee #1

Reviewer comments- RNAseq: *The FPKM > 1 filter that is used here is quite strict. It would not be surprising if most Y transcripts are not picked up at all. Have you tried analyzing the data using DESeq2?*

Authors' response: We appreciate the reviewer's insightful comment. As described in the Methods, we did perform differential expression analysis using DESeq2. To reduce background noise from extremely low-abundance transcripts, we retained only genes with FPKM > 1 in both groups prior to testing. Similar filtering strategies have been used in previous studies (Li et al., PNAS, 2021; Lin et al., Cell Reports, 2024., Han et al, 2021; Sangrithi et al, 2017).

This filtering step excluded only one Y-linked gene, which did not show significant differential expression. The Y chromosome only contains sixteen gene families (Subrini et al., Science, 2025), so this filtering strategy had minimal impact on the analysis. These results further support our conclusion that MSCI remains unaffected in *Spdya*^{CKO} spermatocytes, despite the persistent *full Y-X NH synapsis*.

Reviewer comments- scRNAseq: *The generated UMAPs look quite different from other published datasets. In most published datasets, the differentiating spermatogonia and spermatocyte clusters form a continuous trajectory (example of P10 and adult testis dataset DOI: 10.1126/science.ads6495). In addition, the purpose of the scRNAseq is to distinguish cells that are undergoing MSCI in WT samples and to compare that cluster to the mutant. However, the authors have compared the whole spermatocyte populations, which would include cells in leptoneuma (normal X-Y expression), zygonema (high X-Y expression) and pachynema (low to no expression from early to mid pachynema). To make the comparison valid, it should be carried out only between cells in mid-pachynema, which would require further clustering of the SC cluster (see for list of marker genes, DOI: 10.1038/s41586-022-05547-7, DOI: 10.1038/s41467-019-09182-1).*

Authors' response: We thank the reviewer for the thoughtful comments regarding the scRNA-seq analysis.

We would like to clarify that our single-cell RNA sequencing was performed on synchronized mouse testis samples, as described in the Methods and illustrated in Appendix Figure S1. Testes were collected at RA11.75 (i.e. PD20.75), a time point at which the seminiferous tubules only contain one generation mid-late pachytene spermatocytes, as confirmed by expression of stage-specific marker genes such as *Hspa2*, *Setx*, and *Rsph1*

This synchronization strategy excluded other spermatocyte stages, which explains why our UMAP projections differ from previously published whole-testis datasets that display a continuous spermatogenic trajectory. It also ensures that our analysis specifically reflects the mid-to-late pachytene stage, during which MSCI is fully established.

We hope this explanation adequately addresses the reviewer's concerns.

References

1. Li M, Zheng J, Li G, Lin Z, Li D, Liu D, et al. The male germline-specific protein MAPS is indispensable for pachynema progression and fertility. *Proceedings of the National Academy of Sciences of the United States of America* 2021, 118(8).
2. Lin Z, Li D, Zheng J, Yao C, Liu D, Zhang H, et al. The male pachynema-specific protein MAPS drives phase separation in vitro and regulates sex body formation and chromatin behaviors in vivo. *Cell reports* 2024, 43(1): 113651.
3. Han G, Hong SH, Lee SJ, Hong SP, Cho C (2021) Transcriptome Analysis of Testicular Aging in Mice. *Cells* 10
4. Sangrithi MN, Royo H, Mahadevaiah SK, Ojarikre O, Bhaw L, Sesay A, Peters AH, Stadler M, Turner JM (2017) Non-Canonical and Sexually Dimorphic X Dosage Compensation States in the Mouse and Human Germline. *Developmental cell* 40: 289-301.e283
5. Subrini J, Varsally W, Balsells IB, Bensberg M, Sioutas G, Ojarikre O, Maciulyte V, Gylemo B, Crawley K, Curtis K, de Rooij DG, Turner JMA. Systematic identification of Y-chromosome gene functions in mouse spermatogenesis. *Science*. 2025 Jan 24;387(6732):393-400.

Referee #2

Reviewer comments: I just read the authors' revisions and I was perfectly satisfied by their efforts to address the comments point by point. The quality of the manuscript has significantly improved and I believe is ready for publication.

Authors' response: We sincerely thank the reviewer for the positive feedback and encouraging comments. We truly appreciate your time and effort in reviewing our manuscript and are grateful for your support for its publication.